# The Mla system of diderm Firmicute *Veillonella parvula* reveals an ancestral transenvelope bridge for phospholipid trafficking

Kyrie P. Grasekamp[1], Basile Beaud Benyahia[2,9], Najwa Taib[2,3,9], Bianca Audrain[1,9], Benjamin Bardiaux [4,5,9], Yannick Rossez [6], Nadia Izadi-Pruneyre [4,5], Maylis Lejeune [4,5], Xavier Trivelli [7], Zina Chouit[6], Yann Guerardel [6,8], Jean-Marc Ghigo [1], Simonetta Gribaldo [2] ✉ & Christophe Beloin [1] ✉

*E. coli* and most other diderm bacteria (those with two membranes) have an inner membrane enriched in glycerophospholipids (GPLs) and an asymmetric outer membrane (OM) containing GPLs in its inner leaflet and primarily lipopolysaccharides in its outer leaflet. In *E. coli*, this lipid asymmetry is maintained by the Mla system which consists of six proteins: the OM lipoprotein MlaA extracts GPLs from the outer leaflet, and the periplasmic chaperone MlaC transfers them across the periplasm to the inner membrane complex MlaBDEF. However, GPL trafficking still remains poorly understood, and has only been studied in a handful of model species. Here, we investigate GPL trafficking in *Veillonella parvula*, a diderm Firmicute with an Mla system that lacks MlaA and MlaC, but contains an elongated MlaD. *V. parvula mla* mutants display phenotypes characteristic of disrupted lipid asymmetry which can be suppressed by mutations in *tamB*, supporting that these two systems have opposite GPL trafficking functions across diverse bacterial lineages. Structural modelling and subcellular localisation assays suggest that *V. parvula* MlaD forms a transenvelope bridge, comprising a typical inner membrane-localised MCE domain and, in addition, an outer membrane ß-barrel. Phylogenomic analyses indicate that this elongated MlaD type is widely distributed across diderm bacteria and likely forms part of the ancestral functional core of the Mla system, which would be composed of MlaEFD only.

The diderm bacterial envelope is a complex structure consisting of two membranes: a cytoplasmic - or inner - membrane (IM) composed primarily of glycerophospholipids (GPLs), and an outer membrane (OM) which exhibits lipid asymmetry, comprising GPLs in its inner leaflet and primarily lipopolysaccharide (LPS) molecules in its outer leaflet. This asymmetry manifests as a robust exclusion barrier against a range of antibiotics and host components, such as vancomycin and bile salts[1,2].

Although GPLs are one of the most ubiquitous amphipathic components of the diderm envelope, their transport systems remain poorly understood, and most current information stems from a handful of model bacterial species. Recent studies implicate large AsmA-like proteins, such as TamB and YhdP, in the mediation of anterograde GPL trafficking[3–5], whilst MCE proteins—namely MlaD within the maintenance of lipid asymmetry (Mla) machinery—are proposed to

---

facilitate retrograde GPL trafficking[6,7]. In model diderms such as *E. coli* and *A. baumannii*, the Mla system is composed of six proteins (Fig. 1): MlaBDEF form the IM complex[6,8–10], MlaC is the periplasmic chaperone[6,11], and MlaA is the OM lipoprotein that associates with porin trimers[12]. Despite extensive research of the Mla system within model diderms for over a decade, several mechanistic details, including substrate specificity and even directionality until recently, remain controversial[9,10,13,14].

Here, we investigate GPL trafficking in *Veillonella parvula*, a genetically tractable member of the Firmicutes (Negativicutes) which presents a diderm envelope with LPS[15,16]. *V. parvula* is phylogenetically distant from classical models such as *E. coli* as it belongs to the Terra-bacteria, one of the two large clades into which Bacteria are divided (containing phyla such as Actinobacteria, Cyanobacteria and Candidate Phyla Radiation (CPR)), and whose cell envelopes are largely understudied[16,17]. Remarkably, the Firmicutes is the only phylum so far identified to contain a mixture of both monoderm and diderm clades, offering an ideal platform on which to study both the monoderm-diderm transition[18] and the diversity of OM biogenesis systems[16]. We previously demonstrated that the diderm cell envelope architecture is an ancestral feature, which was already present in both the ancestor of the Firmicutes and the Last Bacterial Common Ancestor (LBCA)[15–18], further supported by a recent study[19]. Studying diderm Firmicutes can therefore shed light on the evolution of ancestral systems in the biogenesis and maintenance of the bacterial OM. Interestingly, the *V. parvula* genome contains an operon—embedded in a large OM biogenesis and maintenance gene cluster[15,18]—with only three homologues of the Mla system: the IM proteins MlaEFD, directly followed by a homologue of the OM efflux protein TolC[20], and TamB (Fig. 1). This arrangement is conserved in all Negativicutes (Supplementary Fig. 1). However, no homologues of the periplasmic chaperone MlaC or the OM lipoprotein MlaA were identifiable at the sequence level, consistent with the fact that no OM lipoproteins or homologues of the Lol system have so far been identified in *V. parvula*[16,21]. This raises the question of how these *V. parvula* Mla homologues are involved in retrograde lipid trafficking, and how they can accomplish this function in the absence of MlaAC.

In this study, we combine phenotypic characterisation, lipid content analysis, structural modelling and evolutionary reconstructions to show that the three-component Mla system of *V. parvula* is involved in GPL trafficking, and is strikingly different to what has been described so far in model diderms; MlaEFD appear to form a functional 'minimal' system, in which a transenvelope MlaD abrogates the requirement for the missing homologues MlaA and MlaC by directly bridging the IM and OM. Further, we show that MlaEFD represent an ancestral core for GPL trafficking that is widely distributed across Bacteria and likely dates back to the LBCA, whereas the presence of MlaABC is an exception, emerging later in the Proteobacteria and other closely related phyla. Finally, and most strikingly, we show that the majority of MlaD sequences across the bacterial kingdom are 'long', revealing that the short MlaD employed by Proteobacteria is atypical. Together, our results uncover novel functional information about GPL trafficking in a non-model organism, shedding light on the

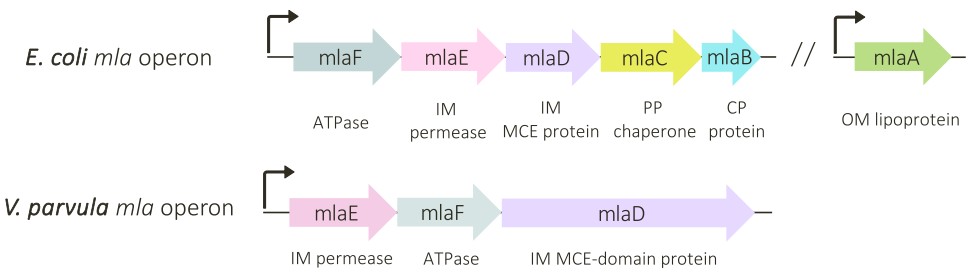

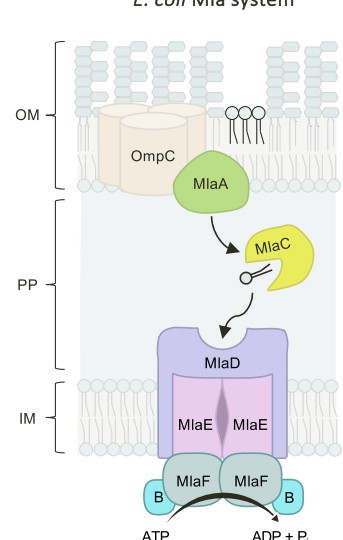

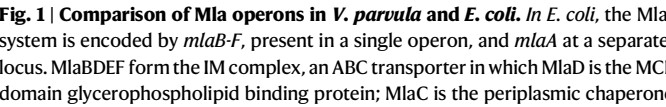

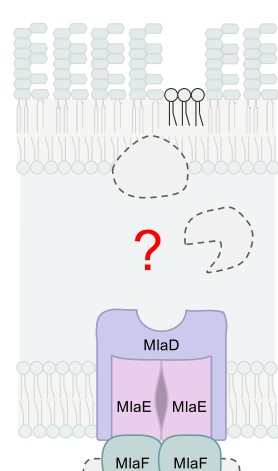

**Fig. 1 | Comparison of Mla operons in *V. parvula* and *E. coli*.** In *E. coli*, the Mla system is encoded by *mlaB-F*, present in a single operon, and *mlaA* at a separate locus. MlaBDEF form the IM complex, an ABC transporter in which MlaD is the MCE-domain glycerophospholipid binding protein; MlaC is the periplasmic chaperone, and MlaA is the OM lipoprotein that associates with OmpC trimers. In *V. parvula*, only homologues of *mlaEFD* were identified. These genes are encoded together in an operon, immediately upstream of a homologue of *tolC* and *tamB*. OM outer membrane, PP periplasm, IM inner membrane, CP cytoplasm.

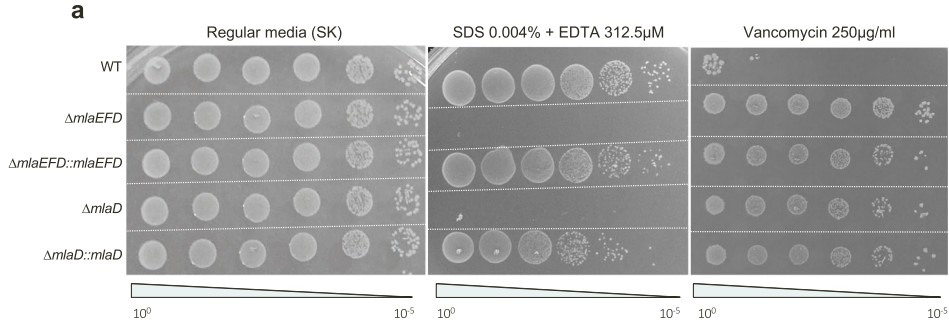

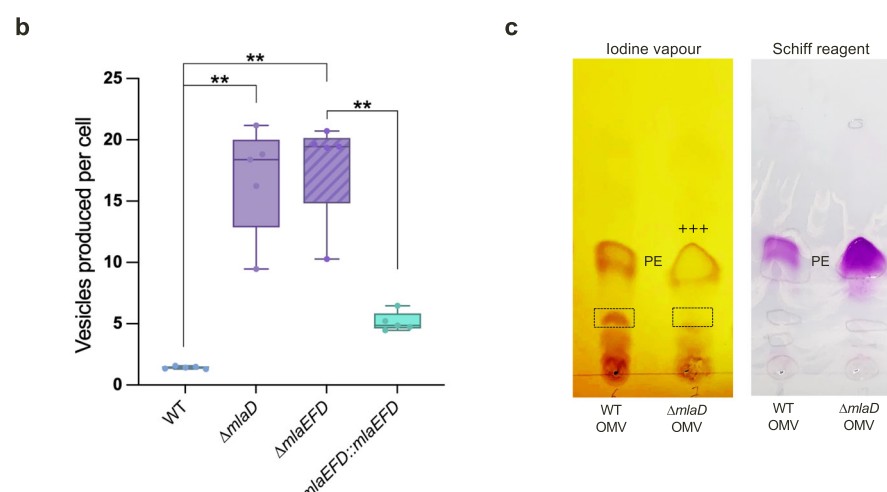

**Fig. 2 | Phenotypic characterisation of *mla* mutants reveals detergent hypersensitivity and hypervesiculation. a** Efficiency of plating assay shows SDS / EDTA sensitivity of *mla* deletion strains. Overnight cultures were adjusted for $OD_{600}$ and serial diluted onto SK media supplemented with SDS / EDTA and vancomycin. Δ*mla* mutants display hypersensitivity to SDS / EDTA, and a strong resistance to vancomycin. Complementation of the detergent hypersensitivity was performed with an aTc-inducible vector pRPF185. **b** Hypervesiculation of *mla* deletion strains. Outer membrane vesicles (OMVs) produced by the WT, Δ*mlaD*, Δ*mlaEFD* and complemented strains were quantified via NanoFCM (see methods) and used to generate a ratio of hypervesiculation. The WT produces a similar number of vesicles to cells, resulting in a ~1:1 ratio, whilst the Δ*mla* deletion strains produce ~17-fold more vesicles than the WT. This phenotype can be significantly rescued, reducing vesicle production to ~5-fold in the Δ*mlaEFD::mlaEFD* strain. 5 biological replicates and 3 technical replicates were tested per strain; *n* = 5; significance calculated by Mann Whitney U, ** corresponds to *p* = 0.0079; two-tailed p-value. Box plot definition: the centre line denotes the median value (50th percentile), while the box contains the 25th to 75th percentiles of dataset. The whiskers mark the 5th and 95th percentiles. **c** Enrichment of phosphatidylethanolamine (PE) in OMVs produced by Δ*mlaD*. Thin layer chromatography (TLC) of lipid extracts from OMVs produced by both the WT and Δ*mlaD* strains suggest a relative enrichment for PE in the vesicles produced by the Δ*mla* mutants, as compared to lipid 2, when stained with iodine vapour (labelled +++, quantified by ImageJ v1.53 over 3 biological replicates). Schiff reagent staining suggests a ~130% increase in the abundance of plasmenyl PE in Δ*mlaD* OMV lipid extracts.

evolution of the Mla system, challenging the assumption that OM biogenesis systems studied in *E. coli* represent the majority of diderm bacteria, and highlighting the diversity within OM biogenesis and maintenance systems.

## Results

### A "minimal" Mla system in *V. parvula* is involved in GPL trafficking

During a screen to identify genes required for the generation and maintenance of the OM barrier in *V. parvula* SKV38 (see methods), we identified a transposon insertion in a homologue of *mlaD*, and a mixed clone containing an insertion in both *mlaE* and a gene of unknown function, suggesting a role for these *mla* genes in envelope biogenesis. To further investigate the function of the *V. parvula* MlaEFD system, we generated the corresponding single and triple *mlaEFD* deletion mutants and assessed their phenotypes.

Consistent with Mla-defective phenotypes of model diderms such as *E. coli* and *A. baumannii*[7,13], all *V. parvula* Δ*mla* strains—with the exception of the *mlaE* mutant that could not be obtained, perhaps due to a toxicity of MlaD and/or MlaF in the absence of MlaE—display hypersensitivity to SDS/EDTA and increased resistance to vancomycin

(Fig. 2a), with otherwise no differences in viability, morphology, biofilm formation or aggregation (Supplementary Fig. 2). Phenotypes were not additive across single and triple mutants, suggesting these three *mlaEFD* genes work together as part of a single system, further implied by the overlapping start and stop codons of *mlaF* and *mlaD*. SDS/EDTA sensitivity could be complemented in both single and triple mutants, and appeared slightly fuller for the triple mutant, suggesting that correct stoichiometry in the expression of all three genes is important for phenotypic rescue. Δ*mlaEFD* and Δ*mlaD* strains also displayed a ~17-fold increase in outer membrane vesicle (OMV) production as compared to the WT (Fig. 2b), consistent with previously described hypervesiculation phenotypes of Δ*mla* strains in *Vibrio cholerae*, *Haemophilus influenzae*, *Bordetella pertussis* and *Neisseria gonorrhoeae*[22-24], which could be significantly rescued via complementation to a ~4-fold increase in OMV production as compared to the WT strain. Together, these phenotypes are indicative of OM remodeling, and possibly a loss of lipid asymmetry[22,25]. To determine if these changes in membrane permeability and stability are due to a change in quantity or composition of LPS, silver-staining was used to assess relative LPS profiles of these *mla* mutants, revealing no observable differences (Supplementary Fig. 2b).

Previous work in model diderms has inferred the function of the Mla system by exploiting other pathways that maintain lipid asymmetry, such as overexpression of the phospholipase A1 encoding gene (*pldA*) to rescue detergent hypersensitivity[7] and radiolabeling with PagP to follow the hepta-acylation of lipid A[7,26]. As homologues of these systems are not present in *V. parvula*, we relied on lipid extraction and comparative thin layer chromatography (TLC) to infer the function of the *mla* genes. However, to understand potential changes in lipid composition due to deletion of the Mla system, we first had to determine the nature of the major lipids present in *V. parvula* SKV38. We identified homologues of genes that may be involved in the biosynthesis of phosphatidylethanolamine (PE), phosphatidylserine (PS) and phosphatidylglycerol (PG). In addition, we identified a homologue of the recently identified hydrogenase/reductase operon required for plasmalogen biosynthesis in *Clostridium perfringens*[27], but encoded within a single gene (Supplementary Fig. 3a). Using TLC, matrix-assisted laser desorption/ionisation quadrupole ion trap time-of-flight (MALDI-QIT-TOF) and nuclear magnetic resonance (NMR) spectroscopy on lipid extracts from the WT, we then revealed that the major lipid species in the envelope of *V. parvula* are PE (~68%) and PG (~20%), along with two other uncharacterised lipids representing a further ~9% and ~3% of the envelope (Supplementary Fig. 3, Supplementary Table 1). NMR analyses also revealed that ~15–35% of each lipid species present in the envelope of *V. parvula* contains a vinyl ether bond (indicative of plasmalogens[27,28]), which could result from the activity of the identified plasmalogen biosynthesis homologue (Supplementary Fig. 4). No cardiolipin was observed via MALDI-QIT-TOF, consistent with the absence of homologues of genes involved in the synthesis of this lipid in the *V. parvula* genome.

Both whole-cell and membrane lipid extracts from WT and Δ*mla* strains were used to perform TLC, however no differences in GPL composition were observed, as previously described in *E. coli*[7] (Supplementary Fig. 5a). We posited that this lack of difference may be due to the observed hypervesiculation of Δ*mla* strains, compensating for OM lipid imbalance by shedding excess envelope material. We therefore purified OMVs produced by both the WT and Δ*mlaD* strains, performed lipid extraction and TLC, and revealed different classes of lipid species by staining with iodine vapour, Schiff reagent and phosphomolybdic acid (PMA). Across all iodine vapour-stained plates, OMVs produced by Δ*mlaD* strains display a relative enrichment for PE when compared to another lipid species that is abundant in WT OMVs (Fig. 2c, Supplementary Fig. 5b). These results suggest that Δ*mla* mutants accumulate PE in their OM, which is subsequently shed from the envelope via excessive vesicle formation. Taken together, this analysis suggests that the minimal MlaEFD system of *V. parvula* is involved in retrograde GPL trafficking.

### Suppressor analysis implicates Mla and TamB in coordination of GPL homeostasis in *V. parvula*

To further strengthen the proposed function of the Mla system in *V. parvula*, we investigated the genes able to suppress detergent hypersensitivity of Δ*mla* strains. By performing random transposon mutagenesis in the Δ*mlaD* background, we identified ten chromosomal insertion mutants able to grow in SK media supplemented with 0.004% SDS and 312.5 μM EDTA. Efficiency of plating revealed that only four of these mutants could significantly rescue the Δ*mlaD* phenotype (Fig. 3a), of which two were of particular interest: sequencing revealed identical Tn insertions at the same position in the downstream homologue of *tamB* (Fig. 3b).

Considering the recent implications of TamB and other large AsmA-like proteins in anterograde GPL trafficking[3,4], it was interesting to identify this gene as a suppressor of Δ*mlaD* detergent sensitivity. Indeed, we had previously identified a spontaneous suppressor of Δ*mlaD* with a SNP encoding a STOP codon in *tamB*, though the phenotype was unstable and reverted (Fig. 3b). We therefore generated

clean, stable deletion mutants to further probe the relationship between these two genes. As with the suppressor screens, loss of *tamB* partially rescues SDS/EDTA sensitivity in Δ*mlaD* (Fig. 3c). Interestingly, the *tamB* mutant presents a permeability phenotype which appears to be the opposite of *mla* mutants, with no change in SDS/EDTA sensitivity and an increased sensitivity to vancomycin, as previously described in *E. coli*[3] (Fig. 3c, Supplementary Fig. 6b). Much like Δ*mlaD*, however, Δ*tamB* also displays a hypervesiculation phenotype, with a ~10-fold increase in vesicle production as compared to the WT strain (Fig. 3d). Strikingly, loss of *tamB* in the Δ*mlaD* background significantly reduces hypervesiculation; the Δ*tamB*Δ*mlaD* mutant produces only ~5-fold more vesicles than the WT (Fig. 3d). This rate of reduction in OMV production is comparable to the phenotype of the fully complemented Δ*mlaEFD::mlaEFD* strain, suggesting that deletion of *tamB* is as effective at rescuing this envelope defect as reintroducing the *mla* genes. The contrasting OM permeability phenotypes of Δ*mlaD* and Δ*tamB*, together with the striking suppression of the Δ*mlaD* OM-related phenotype by the *tamB* mutation, hints to the antagonistic functions of these two genes in envelope biogenesis and maintenance. Overall, these results suggest that MlaEFD and TamB may represent two major systems maintaining GPL homeostasis in the OM of *V. parvula*—likely retrograde and anterograde, respectively.

### Structural modelling suggests a transenvelope Mla system in *V. parvula*

Though phenotypic and suppressor analyses elucidated the functional role of MlaEFD in *V. parvula*, we sought to understand how the predicted structure of the IM MlaEFD complex could facilitate GPL trafficking in the absence of OM and periplasmic components MlaAC. Whilst the *mlaE* and *mlaF* homologues closely resemble those of model diderms such as the Proteobacteria, *mlaD* in *V. parvula* is over double the size of that of *E. coli* (419 amino acids vs 183 amino acids respectively) (Fig. 4a). Both versions of this gene contain a transmembrane domain (TMD) that anchors the protein in the IM, and a conserved MCE domain that binds lipids[6]. However, MlaD in *V. parvula* extends with a long-predicted α-helical region spanning ~150 residues, and a predicted C-terminal β-barrel (Fig. 4a).

AlphaFold2[29] (AF2) was used to predict the structure of this unusually long MlaD protein, but was unable to confidently model the elongated α-helical region, and indeed the positioning of this domain relative to the N- and C-termini of the protein (Supplementary Fig. 7). However, high confidence predictions were achieved for the TMD and MCE domain, and for the C-terminal β-barrel. This C-terminal domain displays characteristic features of a membrane-embedded β-barrel, composed of ten alternating antiparallel β-strands with a hydrophobic and neutral surface for the β-sheets, and with 'girdles' of negatively charged and aromatic residues which flank the top and bottom of the structural model (Supplementary Fig. 8).

Considering all characterised MCE proteins to date form hexamers, we built a structural model of the full-length protein by merging models obtained with AlphaFold-Multimer[29,30] of hexameric configuration of all domains of MlaD$_{Vp}$ with overlapping segments (Fig. 4b). Even when generated without templates, a 2:2:6 stoichiometry of the MlaEFD$_{1-130}$ complex is confidently predicted for *V. parvula*, with high similarity to the resolved structure of the MlaDEF complex from *E. coli*[9]. This MlaEFD$_{Vp}$ model includes both the N-terminal elbow helix in MlaE, and a shorter version of the C-terminal extension of MlaF interacting with the neighbouring MlaF subunit, despite the absence of an MlaB counterpart (Supplementary Fig. 8). Interestingly, for the hexameric α-helical region of MlaD$_{Vp}$, two major models are generated: one with a closed tunnel structure, and one with an open pore or groove with little interaction between the first and last chains of the hexamer (Supplementary Fig. 9, Supplementary Table 2). We suspect that the "open groove" configuration may represent an artefact of the prediction, owing to the way AlphaFold-Multimer concatenates

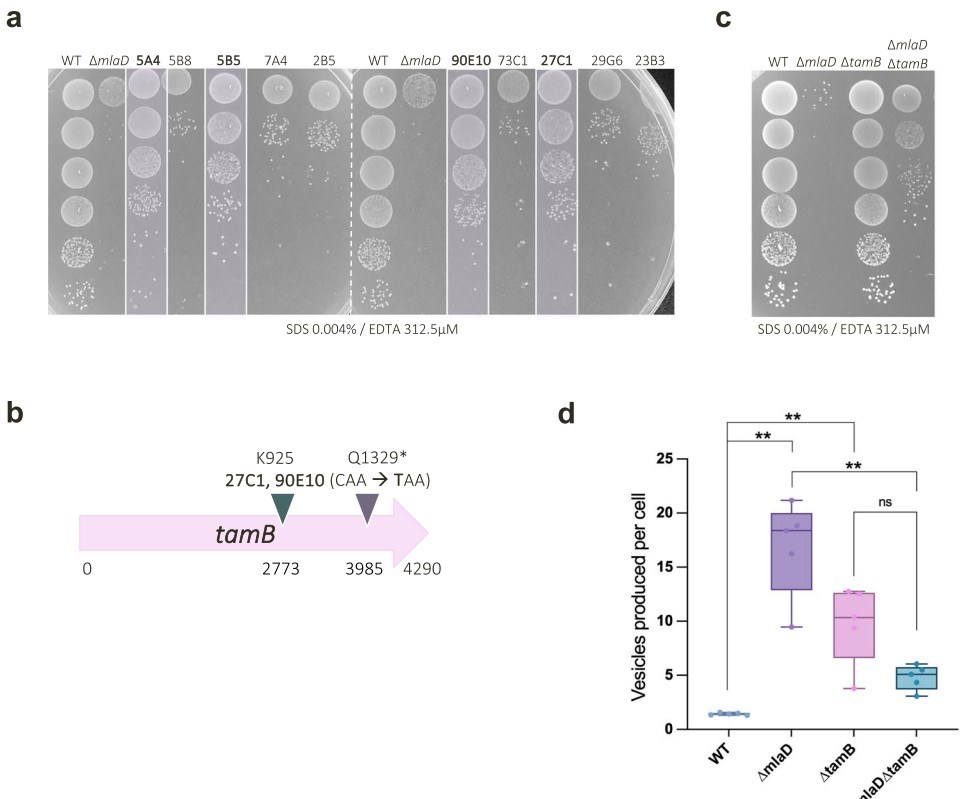

**Fig. 3 | TamB is a suppressor of the Δ*mlaD* OM permeability and hypervesiculation phenotype. a** Extent of phenotypic rescue of Δ*mlaD* across 10 Tn mutants. Efficiency of plating assays on SDS 0.004% + EDTA 312.5 μM were used to determine the extent of phenotypic rescue by random Tn insertions. Only 4 of the 10 Tn insertions could significantly rescue the detergent sensitivity of Δ*mlaD*: 5A4, 5B5, 27C1 and 90E10. **b** Suppressor mutations identified in *tamB*. From both spontaneous suppressor analysis, and random Tn insertion suppression analysis, we identified mutations in a homologue of *tamB*. The two identical Tn insertions were at K925, whilst the SNP in the spontaneous suppressor was mapped to Q1329*, generating a STOP codon (CAA → TAA). **c** OM permeability of WT, *tamB* and *mla* mutants. Clean deletion mutants were generated for Δ*tamB*, Δ*mlaD* and Δ*mlaD*Δ*tamB*, then overnight cultures of these strains and the WT were serially diluted onto SDS / EDTA to assess OM permeability. Δ*tamB* does not display a hypersensitivity to SDS 0.004% / EDTA 312.5 μM, whilst Δ*mlaD* is almost unable to grow at this concentration. Deletion of *tamB* in this Δ*mlaD* background partially rescues the detergent hypersensitivity of Δ*mlaD*. **d** Outer Membrane Vesicle (OMV) production of WT, Δ*tamB* and Δ*mla* mutants. OMV production of WT, Δ*tamB*, Δ*mlaD* and Δ*mlaD*Δ*tamB* strains were analysed via NanoFCM quantification. At least 5 biological replicates and 2-3 technical replicates were tested per strain. Whilst Δ*mlaD* and Δ*tamB* display hypervesiculation, the double mutant (Δ*mlaD*Δ*tamB*) shows a significant reduction in OMV production as compared to Δ*mlaD*; $n = 5$; significance calculated by Mann Whitney U, ** corresponds to $p = 0.0079$; two-tailed p-value. This reduction in hypervesiculation is non-significant between Δ*tamB* and Δ*mlaD*Δ*tamB* ($p = 0.0952$). WT and Δ*mlaD* OMV production data previously shown in Fig. 2 above. Box plot definition: the centre line denotes the median value (50th percentile), while the box contains the 25th to 75th percentiles of dataset. The whiskers mark the 5th and 95th percentiles.

sequences of all chains prior to structure inference. Instead, the closed tunnel model (Supplementary Fig. 9) more closely reflects the resolved crystal structure of PqiB, another MCE-domain protein from *E. coli*, which forms a closed α-helical tunnel on top of three hexameric MCE rings[6], and the recently resolved heterohexamer of Mce1A-1F from *Mycobacterium smegmatis*[31]. Like these resolved structures, the predicted tunnel formed by the α-helices generates a hydrophobic interior of ~13 Å in diameter (Fig. 4c) - supporting the possibility of a role in hydrophobic substrate transport - and a total possible length of ~23.5 nm, compatible with the periplasmic width in *V. parvula* of 24.2 ± 1.8 nm ($n = 26$) as measured by cryo-EM. Further, both the N- and C-termini of the full-length model generate highly hydrophobic bands where both regions are predicted to embed within the IM and OM, respectively. The rest of the structure, which would reside in the aqueous periplasm, displays a hydrophilic exterior (Fig. 4c). It is interesting to note the prediction of a small external cluster of hydrophobic residues at the lowest confidence region of the tunnel – which coincides with the region that forms a groove in the higher confidence model.

From these high-confidence structural predictions of a membrane-embedded C-terminal β-barrel and an IM MCE-domain

hexamer, we hypothesised that MlaD from *V. parvula* may have domains anchored in both the IM and OM, forming a transenvelope bridge. This would remove the necessity for a periplasmic chaperone boat and OM lipoprotein, as MlaD would be capable of connecting the two membranes directly to facilitate transport. To verify this hypothesis experimentally, we localised full-length MlaD and its separate domains by performing membrane fractionation by sedimentation on sucrose density gradients. We used total membrane pellets (containing both IM and OM vesicles) obtained from WT and Δ*mlaD* strains complemented with HA-tagged versions of the TMD-MCE domain and the β-barrel. Localisation of these domains and of the full-length protein was examined via immunoblotting with our developed MlaD antisera, and the separate domains with a commercial anti-HA antibody. Whilst the full-length MlaD was distributed across almost all membrane fractions, the TMD-MCE domain construct was most abundant in fractions F4-5, similar to the localisation profile of the IM control SecA, and the β-barrel was located almost exclusively in the final few OM fractions, F9-11, mirroring the localisation of TolC (Fig. 4d). These results support the hypothesis that MlaD may span the periplasm, with domains anchored in both the IM and OM.

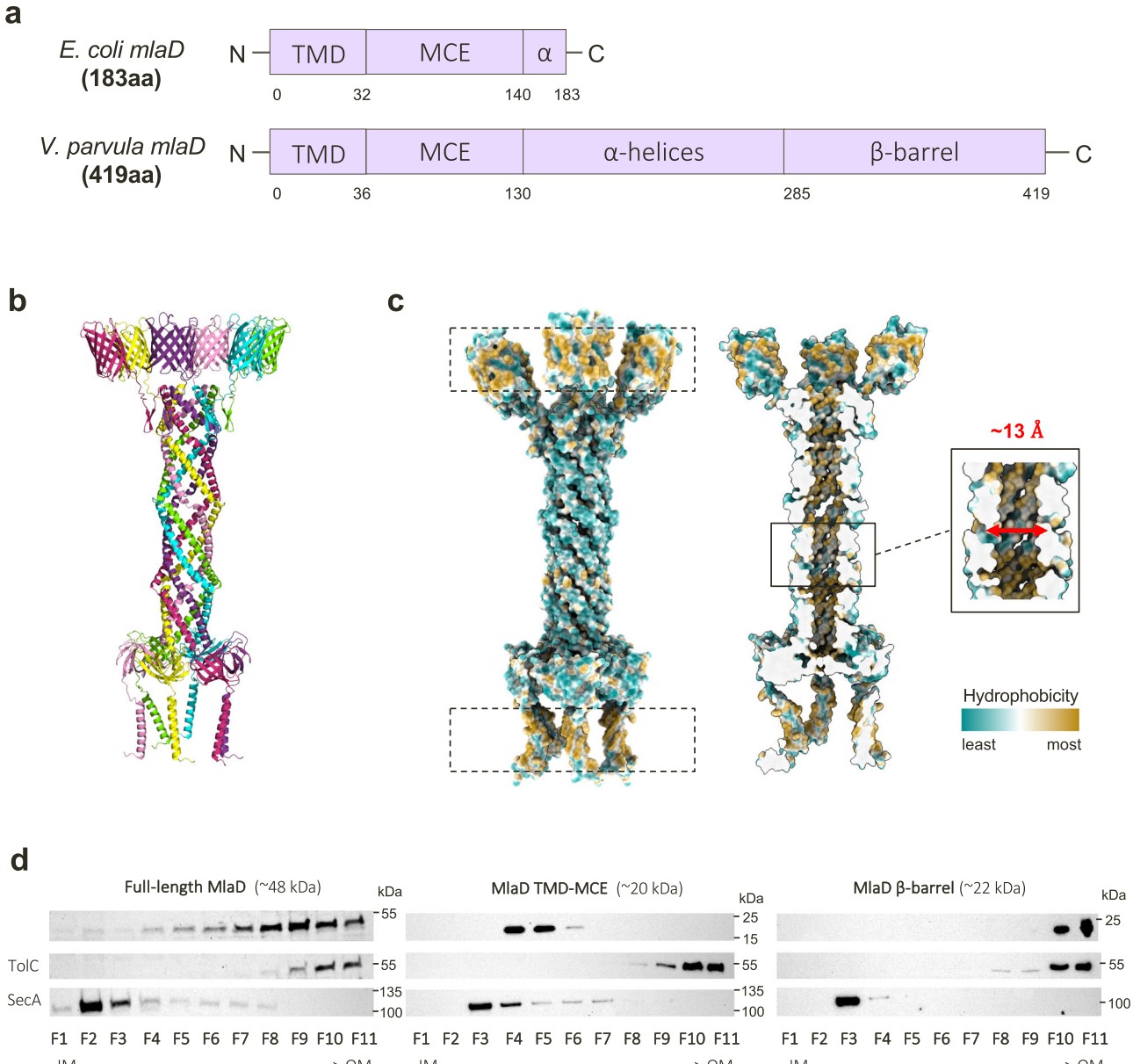

**Fig. 4 | Structural modelling and experimental localisation of a transenvelope MlaD in *V. parvula*. a** Domain comparison of MlaD in *E. coli* and *V. parvula*. MlaD in *E. coli* consists of 183 residues and two major domains: a transmembrane domain (TMD) and an MCE domain, with a small α-helical region at its C-terminus. MlaD in *V. parvula* is 419 residues, consisting of a TMD, MCE domain, extended α-helical region and a β-barrel at its C-terminus. **b** Merged hexameric model of full-length MlaD predicted by AlphaFold2. Separate AF2 models of each domain of MlaD were overlapped to create a merged, full-length hexameric model of MlaD. The Alpha-Fold model of hexameric MlaD can be found in the Supplementary Dataset 3. **c** Hydrophobicity of the full-length hexameric MlaD model predicted by Alpha-Fold2. The full-length MlaD model displays two strikingly hydrophobic regions (dashed boxes), at both the N- and C-termini, which are predicted to both be membrane-embedded domains. The tunnel of this model displays a hydrophobic interior, with a diameter of around ~13 Å. In the left view, 3 β-barrels/TM helices are in front and the 3 others in the back. In the right (cut-away) view, only the 3 β-barrels/TM helices in the back are visible. **d** Subcellular localisation of MlaD domains via sucrose-density gradient membrane fractionation. The WT strain, and Δ*mlaD* complemented with the TMD-MCE domain / β-barrel domain with a C-ter HA-tag, were used for membrane fractionation via sucrose gradient sedimentation. Fractions were visualised with the native MlaD antisera (WT) or anti-HA antibody (MCE / β-barrel strains). The full-length protein is present in all membrane fractions, and most abundant in F7-10. The TMD-MCE construct localises to the IM, most abundant in F4-5, whilst the barrel localises almost exclusively to the final fractions, F10-11, mirroring the localisation profile of the OM control, TolC. Relative fluorescence intensity of these localisation profiles are shown below the immunoblot. The experiments were repeated twice, *n* = 2.

## Phylogenomic analysis reveals an ancestral core of the Mla system

The possibility of a minimal, transenvelope MlaEFD system in *V. parvula* prompted us to explore its potential existence in other bacterial lineages. We therefore carried out an exhaustive search for homologues of the six components (MlaABCDEF) in a local databank containing 1083 genomes, representing all current bacterial phyla. To identify MlaA, MlaC, MlaD, and MlaE homologues, we used the Pfam domains PF04333, PF05494, PF02470, and PF02405, respectively. As MlaB and MlaF belong to the generic STAS-domain and ATPase families, respectively, we used MacSyFinder2[32] to identify them when found in synteny with at least one of the other Mla components in the genome. The distribution of all six components was then mapped onto a reference phylogeny of Bacteria[16] (Fig. 5, Supplementary Fig. 10, Supplementary Dataset 1).

In agreement with the fundamental role of the Mla system in maintaining OM lipid asymmetry, *mla* homologues are largely found in diderm bacteria (Fig. 5). Some diderm phyla in the Terrabacteria lack any identifiable homologues, which we attribute to two main reasons: either because they also lack the ability to synthesise LPS, like Borrelia, Treponema, Thermotogae and Deinococci[17], which would render the Mla system with its current known function in *E. coli* useless; or possibly due to their atypical envelope architectures: the Thermotogae and Candidate phylum Bipolaricaulota have an OM that is detached at the cell poles[33]; members of the phyla Dictyoglomi and Deinococcus-Thermus both form rotund bodies where cell aggregates share a single OM[33,34]; and cryo-electron tomography of a member of Atribacteria (*Atribacter laminatus*) has recently revealed an atypical cell envelope architecture with an intracytoplasmic lipidic bilayer surrounding the nucleoid[35]. Within the three diderm lineages belonging to the Firmicutes[18], Mla components are found in the Negativicutes and a few Limnochordia, whilst they are completely absent from the Halanaerobiales (Supplementary Fig. 11, Supplementary Dataset 1).

More striking is the overall distribution of MlaDEF vs MlaABC; whilst MlaDEF are widely distributed across almost all diderm phyla, with the exception of those mentioned above, the MlaABC components have a much narrower distribution, present mostly within Proteobacteria, and totally absent in diderm Terrabacteria. Notably, the absence of lipoprotein component MlaA corresponds well to the absence of the Lol system within Terrabacteria (Fig. 5, Supplementary Fig. 11, Supplementary Dataset 1). Interestingly, while we found MlaDE proteins in Corynebacteriales (consistent with their function as cholesterol transporters through the mycobacterial outer membrane, as recently shown[31]), we also identified multiple copies of MlaDE homologues in monoderm members of Actinobacteria - mainly the orders Solirubrobacterales, Thermophilales, Pseudonocardiales - and whose function is unknown.

We then focused specifically on MlaD to investigate whether other bacteria possess the longer version, similar to *V. parvula*. Surprisingly, we found that the short MlaD sequences—similar to that of *E. coli* - are restricted to Proteobacteria, whereas 85% of MlaD homologues are of the longer version (corresponding to an average of 374 residues) (Supplementary Figs. 11–13). To understand if these longer MlaD versions have a similar predicted structure to that of *V. parvula*, we screened them for the presence of β-sheet structures using BOCTOPUS2[36] and AlphaFold2[29]. We could indeed infer the presence of a C-terminal β-barrel in 103 sequences, with a conserved fold and physicochemical properties, some containing an additional helical extension (Supplementary Figs. 8h, 9f). These long versions of MlaD with a β-barrel are widely distributed in the diderm Terrabacteria phyla, whilst in the Gracilicutes they are found mainly in some Proteobacteria and closely related phyla (Fig. 5, Supplementary Fig. 12). The β-barrel is completely absent in MlaD sequences from members of the PVC (Planctomycetes, Verrucomicrobia, Chlamydia) and FCB (Fibrobacteres, Chlorobi, Bacteroidetes) groups (Fig. 5,

Supplementary Figs. 11, 12, Supplementary Dataset 1). Interestingly, many of the long MlaD sequences with no identifiable complete β-barrel domain contain a predicted disordered C-terminal region with some β-structure, often short β-hairpins, which may represent remnants or precursors of a β-barrel (Supplementary Fig. 14). For two of these barrel-less sequences, homohexameric predictions generated by AlphaFold revealed a tiny 6-stranded C-terminal barrel, formed by each monomer contributing a single β-strand (Supplementary Fig. 15). A similar conformation was recently structurally characterised in *M. smegmatis*[31], suggesting barrels may be formed through multimerisation, even in some of the seemingly barrel-less MlaD sequences.

Finally, to infer the origin and evolutionary history of the Mla system, we concatenated MlaD and MlaE sequences into a character supermatrix and inferred their phylogeny. Only markers found in a conserved cluster arrangement (MlaDEF and MlaDE) were used, due to the uncertainty of MlaF annotations. The resulting tree is consistent with the reference phylogeny of Bacteria (Supplementary Fig. 12). It shows, in fact, a clear separation between the Terrabacteria and the Gracilicutes (Ultra-Fast Bootstrap=95%) with the monophyly of phyla within Terrabacteria, indicating vertical inheritance. Contrastingly, the evolutionary history of MlaDE within the Gracilicutes appears more complex, involving several duplications and horizontal transfer events (Supplementary Fig. 12).

Together, our phylogenomic analyses strongly indicate that the majority of diderms may employ a transenvelope three-component Mla system for GPL homeostasis composed of MlaEFD, with the long version of MlaD, like the one present in *V. parvula*. The six-component Mla system characterised in Proteobacteria, containing MlaABC, may therefore represent an exception, rather than the rule. Moreover, we show that this core three-component Mla system is ancestral and was present in the last bacterial common ancestor (LBCA), complementing our previous findings that the LBCA was a diderm with LPS[15–18]. Whilst this minimal MlaEFD system was maintained in diderm Terrabacteria and many Gracilicutes, we propose that it underwent progressive complexification within Proteobacteria with the acquisition of additional Mla components, forming the well-characterised Mla system that we recognise in *E. coli* today.

## Discussion

In this work, we investigated a novel Mla system in the non-model diderm Firmicute *Veillonella parvula*, harboring homologues of MlaEFD only, but none of the OM lipoprotein MlaA or the periplasmic chaperone MlaC. This architecture is consistent with our recent in silico studies suggesting that OM biogenesis systems present in *V. parvula* lack OM lipoprotein components typically seen in Proteobacterial models. For example, the *V. parvula* genome encodes a homologue of BamA, but none of the associated OM lipoproteins, BamB-E[17]; its Lpt system lacks the lipoprotein LptE[17]; and a ß-barrel protein (OmpM) is responsible for OM attachment in contrast to the major lipoprotein tethers Lpp, and Pal found in *E. coli*[16]. These bioinformatic findings spark a fundamental question: are these 'missing' components replaced by non-homologous proteins? Or do these systems function without them?

Our results show that MlaEFD likely perform a similar role in *V. parvula* as has been described in model diderms; deletion of these genes results in OM remodeling and a potential loss of lipid asymmetry, indicated by a detergent hypersensitivity and hypervesiculation phenotype, as well as the intriguing atypical phenotype of vancomycin resistance, also described in *E. coli*. As we further develop our toolkit for lipid analyses in our new model, quantifying the enrichment of phosphatidylethanolamine (PE) in Δ*mla* OMV lipid extracts will also support the putative role of Mla in retrograde PE trafficking in *V. parvula*. Considering ~15% of each lipid species present in these bacteria were determined to be in their plasmalogen form, it is likely that

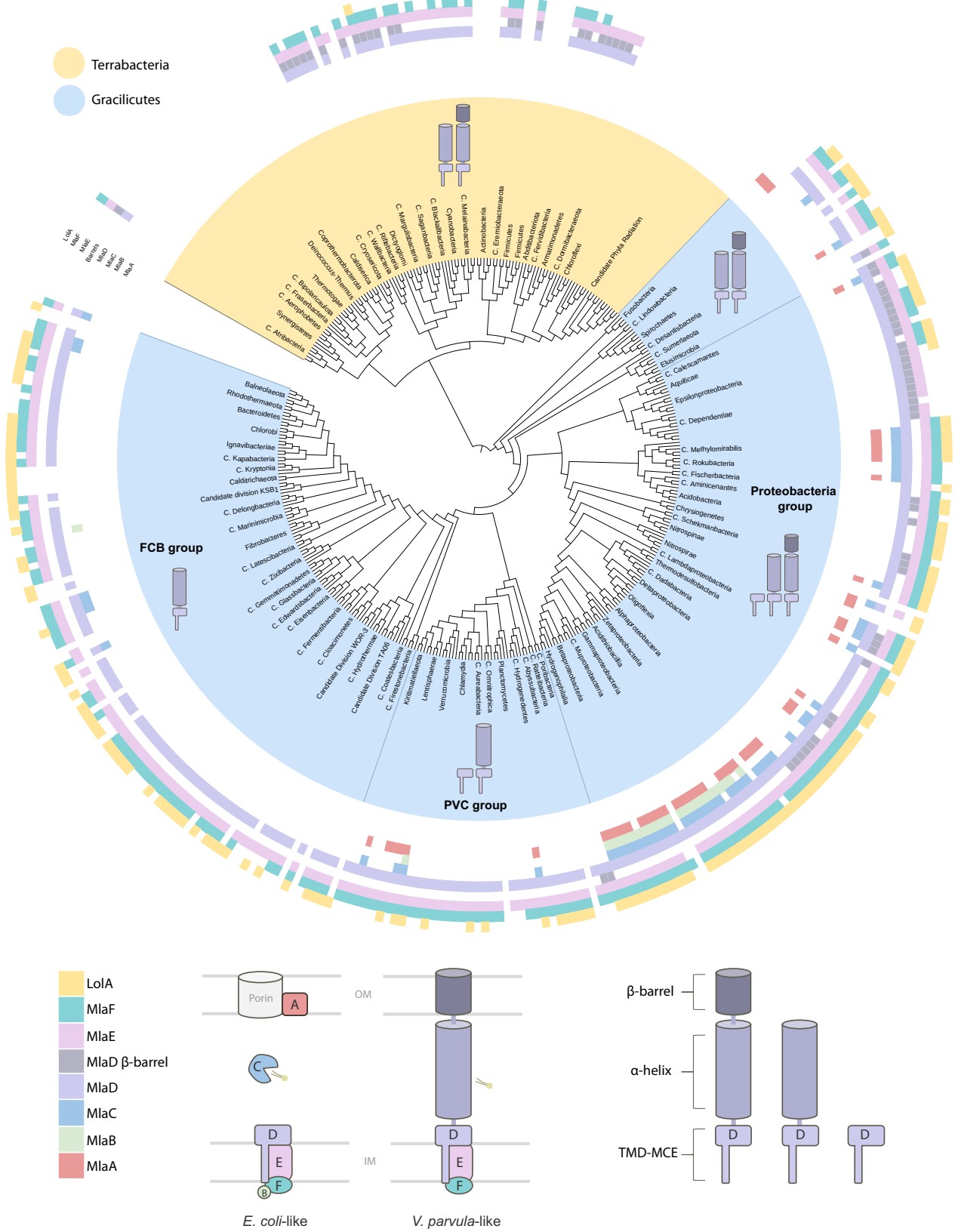

this system shuttles plasmenylethanolamine (PlsPE) as well as typical diacyl PE.

Furthering our understanding of GPL homeostasis in *V. parvula* is the discovery that TamB is a major suppressor of the Δ*mla* phenotype. Given the recent implications of AsmA-like proteins in anterograde GPL trafficking[3–5], it is interesting to discover that this TamB protein

likely performs a similar role in *V. parvula*. These findings, in combination with the recent literature, strongly advocate for the antagonistic roles of TamB and MlaEFD in maintaining the OM of *V. parvula*, performing anterograde and retrograde GPL trafficking respectively. However, as the double Δ*mlaD*Δ*tamB* mutant is viable, we propose that there are as yet undiscovered systems that perform lipid

**Fig. 5 | Taxonomic distribution of MlaABCDEF and LolA across Tree of Bacteria.** The distribution of all six Mla components, and of LolA as a proxy for the Lol system, was mapped onto a reference phylogeny of Bacteria taken from[16] (for a more detailed version of the tree, see Supplementary Fig. 11). In the inner ring of the tree, Gracilicutes are represented by light blue and Terrabacteria in yellow. On the outer ring of the tree, each colored block corresponds to the presence or absence of the six Mla components, A-F, and of LolA (represented by the schematic beneath the tree, left). Two purple blocks are used to represent MlaD; the first light purple block represents presence / absence of the gene itself, and the second dark purple block represents the presence / absence of the β-barrel domain. From this tree, we can see MlaEFD are widely distributed across the bacteria kingdom and were likely present in the LBCA. In contrast, MlaABC are sparsely distributed and restricted mostly to the Proteobacteria, suggesting these components evolved later (and notably the lipoprotein MlaA with the emergence of the Lol system). We also see that, in general, MlaD versions with a β-barrel tend to co-occur with the absence of MlaA and MlaC. The schematic below the tree (right hand side) highlights the three major domains which can be found in MlaD sequences: the transmembrane domain (TMD) and MCE domain, the elongated α-helical region, and the C-terminal β-barrel. The distribution of these 3 'types' of MlaD are represented on the tree, corresponding to which clades contain these MlaD structures. Elongated versions of MlaD are widely distributed across diderm phyla. (Yellow = Terrabacteria; Blue = Gracilicutes). Tree generated using custom-made scripts and iTOL[65]. For detailed species names, see Supplementary Fig. 11.

trafficking in *V. parvula*. Indeed, one other *asmA*-like protein exists in *V. parvula*, and further work is required to elucidate its functional role, if any, in the context of OM homeostasis.

Complementing previous findings that the LBCA was likely a diderm with LPS[15–19], our phylogenomic analysis suggests that the Mla system is also an ancestral feature. We posit that the Mla-based mechanism of 'delousing' the outer leaflet of GPLs via retrograde trafficking is an ancient process for the maintenance of this essential barrier, consistent with the fact that most bacteria which have lost LPS have also lost the Mla system[17]. The existence of some phyla which both possess LPS and lack the Mla system[17] suggests either that they have evolved an alternative retrograde transporter for maintaining lipid asymmetry, or that they rely on degradative mechanisms rather than those of transport, such as PldA hydrolysis as described in *E. coli*[37] or the recently discovered MlaY protein in *P. aeruginosa*[38].

Structural modelling and subcellular localisation helped us to understand how just three *mla* genes may perform the role of GPL trafficking in *V. parvula*; MlaEFD likely form a minimal, periplasm-spanning complex, removing the requirement for the 'missing' MlaAC components by directly connecting the IM and OM with a transenvelope bridge. However, future work is required to understand the stoichiometry of this protein complex, and to further explore the structure of MCE proteins across diverse bacterial lineages. Interestingly, MCE proteins in *E. coli*[6], *A. baumannii*[10] and recently *M. smegmatis*[31] have been shown to form both hetero- and homo-hexamers, and for two of these structures (PqiB and Mce1A-1F), the hexameric nature of the α-helical region results in a closed tunnel that spans the periplasm, with a hydrophobic interior. These studies suggest the tunnel structures could facilitate the transport of a hydrophobic substrate and, from homology, conservation and phenotypic characterisation, the proposed substrate is a glycerophospholipid (or in the case of *M. tuberculosis*, cholesterol).

So far, the only structurally characterised protein that performs retrograde GPL trafficking is MlaC, the periplasmic chaperone that can shield the acyl chains of a phospholipid whilst the head portion remains solvent-exposed[6,11]. This chaperone-based transport is a common theme across diderm OM biogenesis systems, from LolA shielding the tails of nascent lipoproteins[39,40] to Skp / DegP and SurA protecting nascent polypeptides for delivery to the BAM system[41]. However, our phylogenomic analysis shows that the MlaC chaperone – along with MlaA and MlaB - is only sparsely distributed in diderm bacteria. This suggests that the majority of diderms perform retrograde GPL trafficking in a very different way from *E. coli*. Considering that the long version of MlaD is actually the most common one, and that the short *E. coli*-like version of MlaD is almost entirely restricted to the Proteobacteria, it is possible that periplasm-spanning components may be a widespread method of GPL transport in many diderms. Whilst this challenges the paradigm set by extensive research in model diderms such as *E. coli* and *A. baumannii*, it is also supported by recent work in these organisms; evidence now implicates two large, periplasm-spanning AsmA-like proteins, TamB and YhdP, in antero-grade lipid trafficking[3–5]. In these models, large open β-sheet

structures create a hydrophobic groove that is expected to shield the acyl tails of phospholipids in a way analogous to LPS transport by the Lpt system, which employs a β-jellyroll fold, leaving the hydrophilic portion of the lipid substrate solvent-exposed[3,42–44]. Unexpectedly, two different models were generated for the tunnel region of MlaD in *V. parvula:* one with an open groove, and one closed. It is tempting to speculate that this open groove, that winds around the outside of the α-helical structure, could support the tails of the PLs in a 'helter skel-ter'-like fashion, allowing the heads of the phospholipids to remain free in the aqueous environment. However, it is unlikely that six identical α-helices could form a single asymmetrical groove in its structure and, further, the hydrophobic grooves described above are composed of β-sheet structures. From the literature so far, the closed tunnel model is better supported.

Structural characterisation of *V. parvula* MlaD should help to decipher both the stoichiometry and arrangement of the MlaD com-plex, and its mechanism of GPL transport. This holds for the β-barrel domain which could be reminiscent of the OmpC-trimers required for the functioning of the Mla system in *E. coli*[12,45,46]. Indeed, the presence of a β-barrel for substrate translocation across the OM is a common theme across other OM biogenesis systems, such as the requirement for LptD to translocate LPS to the outer leaflet of the OM[47]. However, the predicted MlaD β-barrel is too narrow to allow the passage of GPL, and a hexameric ring of β-barrels is yet to be described - and seems unlikely from AlphaFold predictions (Supplementary Fig. 16). We posit the possibility that several of the β-barrels could fuse to form a trimeric conformation, similar to the trimeric form of TolC that results in a single, larger β-barrel[20].

Importantly, a C-terminal β-barrel was predicted in only ~10% of MlaD sequences, in both diderm Terrabacteria and a clade of Pro-teobacteria. In the remaining MlaD homologues, we predicted a variety of structures: some proteins contain disordered loops with no β-structures, whilst others contain a few β-hairpins in amongst dis-ordered loops. While we cannot speculate at this stage on the order of the evolutionary events that led to this range of disordered, barrel-less MlaD proteins, it is interesting to consider that these unstruc-tured regions could represent either the remnants or even pre-cursors of a β-barrel. Considering the majority of long MlaD sequences are barrel-less, we could speculate that this structure is not important for the putative lipid importer function across bac-teria. However, the recently resolved structure of *M. smegmatis* Mce1A-1F, which do not contain complete C-terminal β-barrels - shows that each monomer contributes a single β-strand, forming a six-stranded β-barrel as a heterohexamer[31]. Perhaps this structural organisation is repeated in other long MlaD sequences without complete barrel structures (as we predicted for some of them using AlphaFold), or there are as yet unidentified proteins necessary to facilitate barrel folding, suggesting a functional convergence across diverse bacterial lineages. The function of these barrels remains to be determined, and may differ across species; perhaps this domain is required for stability or anchoring in some bacteria, and for substrate passage in others.

Regardless of the role of the β-barrel domain, or whether the extended MlaD in *V. parvula* is capable of transporting GPLs via a winding fairground slide or a direct tunnel, the implications are the same: this mechanism is one of higher throughput than a chaperone system. Indeed, this is an important element to consider in the argument regarding the directionality of the Mla system. Whilst a wealth of studies support the function of Mla in retrograde GPL trafficking[7,9,13,48], and several studies argue for the anterograde direction[10,14,49,50], efficiency is a key factor to consider. In the case of anterograde GPL trafficking to facilitate cell growth, GPL flow would have to be enormously high throughput, considering the sheer volume of GPLs required to build a bilayer. In these cases, large periplasm-spanning AsmA-like proteins, which can transport several GPLs at once, would likely facilitate the high flow rate required. On the other hand, a chaperone-based system which can only offer a 1:1 ratio of protein:GPL could not accommodate this high flow rate, and likely stands as a fine-tuning mechanism, which fits with the proposed function of the Mla system, i.e. to remove surface-exposed GPLs from the OM to restore lipid asymmetry. This idea is mirrored by the Lpt system that has evolved to transport LPS into the outer leaflet of the OM, which requires a transenvelope bridge - and not singular protein chaperones. We do not yet understand where the long and likely transenvelope MlaD fits into these scenarios; as MCE proteins are primarily implicated in importer functions, would long MlaD provide high-throughput retrograde GPL trafficking? Or could this process be bi-directional with the same transporter? Perhaps accessory proteins are necessary to control and facilitate the functioning of these transenvelope structures. Intriguingly, as the long MlaD version is widely distributed across bacteria, whatever role this protein performs is likely common across a wide diversity of lineages, waiting to be uncovered.

To conclude, our results reinforce the notion that the paradigms set by extensive studies within the Proteobacteria can be challenged by using non-model and phylogenetically distant bacteria. Research within diderm models such as *E. coli*, *P. aeruginosa* or *A. baumannii* is crucial for our understanding of the functioning of some pathogens. However, learnings from fundamental microbiology – including those from commensals such as *V. parvula* – are often transferable, and may help us fill in the gaps regarding the structure, function and diversity of OM biogenesis and maintenance systems across the Bacterial tree of life.

## Methods

### Bacterial strains and growth conditions
*Veillonella parvula* SKV38 was grown in SK medium (10 g/L tryptone [Difco], 10 g/L yeast extract [Difco], 0.4 g/L disodium phosphate, 2 g/L sodium chloride, and 10 ml/L 60% [wt/vol] sodium DL-lactate; described in ref. 51. Cultures were incubated at 37 °C in anaerobic conditions, either in anaerobic bags (GENbag anaero; bioMérieux no. 45534) or in a C400M Ruskinn anaerobic-microaerophilic station. *Escherichia coli* was grown in lysogeny broth (LB) (Corning) medium under aerobic conditions at 37 °C. Where required, *V. parvula* cultures were supplemented with 20 mg/L chloramphenicol (Cm), 200 mg/L erythromycin (Ery), or 2.5 mg/L tetracycline (Tc), and *E. coli* cultures were supplemented with 100 mg/L ampicillin (Amp). 100 μg/L anhydrotetracycline (aTc) was added to induce the P$_{tet}$ promoter and 1 mM isopropyl-β-d-thiogalactopyranoside (IPTG) was added to induce the T7 promoter. All chemicals were purchased from Sigma-Aldrich unless stated otherwise. All chemicals were purchased from Sigma-Aldrich unless stated otherwise. Primers and strains used in this chapter are listed in Supplementary Tables 3, 4.

### Generation and Screening of Transposon mutagenesis library
A random transposon mutagenesis library was generated in *V. parvula* SKV38 WT, or the Δ*mlaD* mutant, using pRPF215, a plasmid previously used in *Clostridium difficile* that contains an aTc-inducible transposase

and mariner-based transposon (Addgene 106377)[52]. pRPF215 was transformed into *V. parvula* SKV38 by natural transformation and selected on SK agar supplemented with Cm (20 μg/mL). Several independent overnight cultures of *V. parvula* SKV38-pRPF215 were diluted to OD$_{600}$ = 0.1 in SK medium supplemented with aTc (100 ng/μL) and incubated for 5 h to induce the transposase. Following induction, cultures were diluted and plated onto SK supplemented with erythromycin (200ug/μL) and aTc (100 ng/μL) for selection, then incubated in anaerobic conditions for 48 h. Resultant colonies were used to inoculate Greiner Bio-one polystyrene flat-bottom 96-well plates (655101) in SK supplemented with either Ery and aTc or Cm to confirm both the presence of the transposon and the loss of pRPF215. After 24 h incubation, aliquots were taken from each well to inoculate fresh 96-well plates filled with SK and SK supplemented with detergents and antibiotics for the screening process. Original transposon mutagenesis plates were stored in 15% glycerol at −80 °C, whilst the screening plates were incubated for 24 h then assessed for growth and OM permeability defects. Mutants of interest were tested for stability of phenotype, then harvested for genomic DNA using the Wizard® Genomic DNA Purification Kit (Promega). Genomic DNA was sent for whole-genome sequencing at the Mutualized Platform for Microbiology at Institut Pasteur.

### Efficiency of plating / serial dilution assays
Sensibility to different detergents was performed on SK agar plates supplemented with detergents and antibiotics at different concentrations. Serial dilutions of strains ($10^0 – 10^{-5}$) were inoculated with a starting OD$_{600nm}$ = 0.5. Plates were incubated in anaerobic conditions for 48 h and *cfu* were calculated.

### Chromosomal mutagenesis by allelic exchange
To generate deletion strains of *V. parvula* SKV38, site-directed mutagenesis was performed as previously described[51]. Briefly, 1 kb regions upstream and downstream of the target sequence were PCR amplified using Phusion Flash high-fidelity PCR master mix (Thermo Scientific, F548). For the selection process, three main resistance cassettes were PCR amplified with overlapping primers for the upstream and downstream regions of the target gene: the *V. atypica* tetracycline resistance cassette (*tetM* in pBSJL2), the *C. difficile* chloramphenicol resistance cassette (*catP* in pRPF185; Addgene 106367[49] and the *Saccharopolyspora erythraea* erythromycin resistance cassette *(ermE)*. PCR products were ligated to generate deletion cassettes via Gibson cloning or used as templates in a second PCR reaction to generate linear dsDNA with the resistance cassette flanked by the upstream and downstream sequences. These deletion constructs were transformed into *V. parvula* by natural transformation (detailed below), and its genomic integration was selected by plating on tetracycline, chloramphenicol or erythromycin-supplemented medium. Positive candidates were further confirmed by PCR and sequencing.

### Natural transformation
Recipient strains were grown on SK agar for 48 h in anaerobic conditions at 37 °C. Biomass was resuspended in 1 ml SK medium adjusted to OD$_{600}$ = 0.4–0.8. 10 μL aliquots were spotted onto SK agar, then 0.5–1 μg plasmid or 75–200 ng/μL linear double-stranded DNA (dsDNA) PCR product was added, using distilled water as a negative control. Following 48 h incubation, the biomass was resuspended fresh SK medium, plated onto SK agar supplemented with the relevant antibiotic, and incubated for a further 48 h. Colonies were streaked onto fresh selective plates, and correct integration of the construct was confirmed by PCR and sequencing.

### Growth kinetics
Overnight cultures were diluted to 0.05 OD$_{600}$ in 200 μL SK that had previously been incubated in anaerobic conditions overnight to

remove dissolved oxygen in Greiner flat-bottom 96-well plates. To maintain anaerobic conditions, a plastic adhesive film (adhesive sealing sheet, Thermo Scientific, AB0558) was used to seal the plate whilst inside the anaerobic station. The sealed plates were then incubated in a TECAN Infinite M200 Pro spectrophotometer for 24 h at 37 °C. $OD_{600}$ was measured every 30 m after 900 seconds orbital shaking of 2 mm amplitude.

### LPS silver staining (AgNO₃)
**Sample preparation.** Overnight cultures were concentrated to $OD_{600} = 10$ in Tricine Sample Buffer (BioRad 1610739: 200 mM Tris-HCl, pH 6.8, 40% glycerol, 2% SDS, 0.04% Coomassie Blue) and incubated at 100 °C for 10 min. After cooling to room temperature, Proteinase K was added to a final concentration of 1 mg/mL and incubated at 37 °C for 1 h. 10 µL of samples were loaded onto tris-tricine gels (BioRad 4563063: 16.5% Mini-PROTEAN® Tris-Tricine Gel) and migrated in TTS buffer (BioRad 161-0744: 100 mM Tris-HCl pH 8.3, 100 mM Tricine, 0.1% SDS) for 2 h at 50 mA.

**Staining.** To visualise LPS profiles after gel electrophoresis separation, all development solutions were prepared immediately before use. Gels were fixed in ethanol 30%, acetic acid 10% for 1 h, then washed 3 times in dH₂O. Gels were incubated for 10 m with metaperiodate 0.7%, again washed 3 times in dH₂O, and incubated in thiosulfate 0.02% for 1 m. Following a further 3 washes in dH₂O, gels were incubated in AgNO₃ 25 mM for 10-15 m, washed in dH₂O for 15 s, then incubated in development solution (K₂CO₃ 35 mg/ml, formaldehyde 0.03% (w/v), thiosulfate 0.00125%) until bands appeared. To stop the reaction, gels were finally incubated in 4% Tris Base, 2% acetic acid for 30 min, then imaged.

### Spontaneous suppressor selection
Spontaneous suppressor mutations were selected by passaging overnight cultures of Δ*mlaEFD* and Δ*mlaD*, either in liquid media or on agar, supplemented with SDS 0.004% and EDTA 312.5 µM, until growth occurred. Colonies were restreaked onto the same SDS / EDTA concentration, or directly grown in a liquid culture overnight, then genomic DNA was harvested and sent for whole genome sequencing to identify the corresponding SNPs that enabled growth of the mutants in SDS / EDTA.

### Lipid extraction and thin layer chromatography (TLC)
**Sample preparation (lipid extraction).** To extract lipids, typical Bligh-Dyer extraction methods were used: briefly, overnight cultures were pelleted and concentrated to OD600 = 15 in 1 mL of distilled water, then incubated on ice for 5–10 min. Using glass pipettes, a monophasic mixture of 4 mL methanol and 2 mL chloroform were added to the resuspended cells and vortexed vigorously for 5 min. Following a further 10 min incubation on ice, 4 ml chloroform and 2 ml of distilled water were added and samples were vortexed before centrifugation at 1500x *g* for 5 min. This centrifugation step allowed the separation of the two phases: the upper water phase, containing solutes, ions, sugars, DNA and RNA, and the lower chloroform phase, containing the lipids. The upper phase was discarded then 1 mL sodium chloride solution (0.5 M) was added and the samples were vortexed. Following a second centrifugation at 1500 x *g* for 5 min, the lower phase was carefully extracted using glass micropipettes and dried in a vacuum centrifuge or under a flow of nitrogen gas (N₂). If the initial purity was not satisfactory, a secondary sodium chloride step (and centrifugation) could be repeated. Dried lipid extracts were weighed and resuspended in the corresponding volume of chloroform.

**Thin layer chromatography (TLC) and staining.** To visualise lipids, TLC plates (Silica gel 60, nonfluorescent, 0.25 mm thick, Merck Millipore) were cut to a width that allowed ~1 cm between each lipid

sample. Plates were prerun in acetone before use and dried at 100 °C for 30 min. 10 µL of each sample was loaded with 5 µL glass microcapillary tubes (Sigma Aldrich) 1 cm from the bottom of the TLC plate. The plate was then placed in a solvent system optimised for the separation of phospholipids by head group polarity: 65:25:4 (v/v/v) or 80:15:2.5 chloroform/methanol/water. For separation of lipids from OMVs, we used the ratio of 80:20:2.5. When the solvent front reached ~2 cm from the top of the plate, the plate was removed from the solvent system and dried at 100 °C for 30 m or air-dried for 10 min. To reveal phospholipid species, the dried plate was incubated in a glass tank saturated with iodine vapour from iodine crystals until lipid spots were clearly visible (~5 min). After imaging this stained plate, the iodine was allowed to decolourise and the same plate was re-stained with phosphomolybdic acid (PMA, 10%), followed by vigorous heating to develop spots, or Schiff reagent (Sigma-Aldrich). Amines were detected by spraying TLC plates with 0.2 g Ninhydrin (Sigma-Aldrich) dissolved in 100 ml ethanol, followed by vigorous heating to develop spots. For the relative quantification of each lipid class, ImageJv1.53 was used.

### Lipid identification
**Nuclear magnetic resonance (NMR).** After labile proton to deuteron exchange with CD₃OD, phospholipids were dried under vacuum and dissolved in a mixture of CDCl₃/CD₃OD (2:1, v/v). The samples were then introduced into a 3 mm NMR tube. Solution NMR recordings were conducted at 10 or 20 °C on Bruker AVANCE NEO 400 and 900 spectrometers equipped with a 5 mm TBI and 5 mm cryo-TCI probe, respectively. Standard experiments were run: 1D-¹H, 2D-¹H-COSY, 2D-¹H-TOCSY, 2D-¹H-ROESY, 2D-¹H-¹³C-HSQC-DEPT, 2D-¹H-¹³C-HSQC-TOCSY. After acquisition and phase correction, chemical shifts calibration on CHD₂OD signals were performed for ¹H (3.31 ppm) and ¹³C (49.29 ppm). The data were analysed using the TopSpin software (Bruker).

**Mass spectrometry (MS).** MALDI-QIT-TOF Shimadzu AXIMA Resonance mass spectrometer (Shimadzu Europe, Manchester, UK) in the positive mode was used to identify the different lipids. All lipids were dissolved in CHCl₃/CH₃OH (1:1, v/v) and mixed with the same volume (10 µL) of 2,4,6-trihydroxyacetophenone (THAP) dissolved in methanol.

### Outer membrane vesicle (OMV) extraction and purification
Overnight cultures were used to generate $OD_{600}$-adjusted 1 litre cultures of the relevant *V. parvula* strains (WT SKV38 and Δ*mla* mutants). Bacterial cells were pelleted by centrifugation at 5000× *g* for 10 min, and supernatants were passed through 0.45 µm and 0.22 µm pore-size filters to remove any leftover cells or debris (Sarstedt). The filtered supernatant was then concentrated using Centricon Plus-70 centrifugal filters (100 kDa cutoff, Millipore) centrifuged at 3500× *g* 4 °C for 30 min, 50 mL at a time. ~3 mL of concentrated OMV solution was purified from 1 litre of supernatant. This OMV solution was then pelleted by ultracentrifugation at 100,000× *g* for 3 h at 4 °C (Beckman Coulter Optima L-80 XP Ultracentrifuge, Type 50.2 Ti rotor). The mass of the OMV pellets was recorded and used for subsequent lipid extraction and analysis via Thin Layer Chromatography (TLC). Purified OMVs were analysed via NanoFCM confirming an average size of ~60 nm, matching the average size of vesicles in whole cell cultures observed via both NanoFCM and cryo-electron tomography (Supplementary Fig. 7). These observations of OMV preparations are compatible with the absence of cell lysis as also observed in *cfu* counting experiments of WT and Δ*mlaD* strains (see Fig. 2, Supplementary Figs. 2 and 6).

### OMV quantification
Overnight cultures were adjusted for $OD_{600}$ and diluted 1/25 and 1/50 in filtered PBS solution. The size distribution and particle concentration

within the samples were analysed by nano-flow cytometry (nFCM) (NanoFCM, Inc., Xiamen, China). First, the instrument was calibrated for particle concentration using 200 nm PE and AF488 fluorophore-conjugated polystyrene beads, and for size distribution using Silica Nanosphere Cocktail (NanoFCM, Inc., S16M-Exo and S17M-MV). The flow rate and side scattering intensity were then calculated based on the calibration curve, and used to infer the size and concentration of the large and small events present in each sample using the NanoFCM software (NanoFCM Profession V2.0). All samples were diluted to ensure the particle count fell within the optimal range of 2000–12,000/min, and all particles that passed by the detector during time intervals of 60 s were recorded for each test.

### Membrane fractionation by sucrose gradient sedimentation

1.5 L overnight cultures were pelleted at 5000× *g* for 15 min and resuspended in -10 mL HEPES buffer. Resuspended pellets were supplemented with 100 μL benzonase and a small amount of lysozyme prior to lysis by two passages through a high-pressure French press (French Press G-M, Glen Mills) homogenizer at 20,000 psi. Following lysis, cell debris was pelleted at 15,000× *g* for 90 min. Resulting supernatant was spun in an ultracentrifuge (Beckman Coulter Optima L-80 XP Ultracentrifuge, Type 50.2 Ti rotor) at 35,000× *g* for 90 min to obtain the membrane pellet, containing both IM and OM vesicles. Membrane pellets were resuspended in 750 μl of HEPES buffer using a Dounce homogeniser, then 500 μL was loaded onto a discontinuous sucrose gradient comprising a bottom 2 mL layer of 51%, a 3 ml layer of 37%, a 4 mL layer of 31%, and a 2.5 mL top layer of 23% (w/w) sucrose. Sucrose gradient columns were centrifuged at 35,000× *g* for 40 h (Beckman Coulter Optima L-80 XP Ultracentrifuge, Type SW41 Ti rotor). At the end of the sucrose gradient sedimentation, 1 mL by 1 mL was extracted from the top of the gradient column to the bottom, resulting in 1 ml membrane fractions. All fractions were run via SDS-PAGE and resulting gels were used for immunoblotting to localise each protein. Full immunoblots from Fig. 4d (uncropped) are provided in the source data.

### Immunoblotting

Membrane fractions obtained from sucrose gradient sedimentation were denatured using Laemmli buffer (Bio-Rad #1610747) and incubated at 95 °C for 5 min. Samples were run on TGX 4–15% gradient gels (Bio-Rad) at 170 V for 40 min, then resolved proteins were transferred onto 0.2-μm nitrocellulose membranes and stained with the following antibodies: i/ for the localisation of full-length native MlaD from WT membrane fractions, proteins were stained with our own generated primary polyclonal rabbit antibody raised against full-length MlaD used at 1:4000 dilution (CovalAb), then stained with a commercial secondary goat anti-rabbit antibody coupled to HRP (Abcam #98431) at 1:10,000 dilution;

ii/ for the localisation of the MCE-TMD and ß-barrel constructs, membrane fractions obtained from Δ*mlaD* strains were stained with a commercial primary anti-HA-tag polyclonal immunoglobulin coupled to HRP (Novus Biologicals, ref. NB600-363) at a 1:4000 dilution; iii/ for the localisation of the IM and OM controls, SecA and TolC respectively, polyclonal rabbit antisera generated against both of these proteins from *E. coli* were used at a 1:8000 dilution, gifted from Dr Philippe Delepelaire.

### Cryo-electron tomography (Cryo-ET)

A solution of bovine serum albumin–gold tracer (Aurion) containing 10-nm-diameter colloidal gold particles was added to a fresh culture of *V. parvula* with a final ratio of 3:1. A small amount of the sample was applied to the front (4 μl) and the back (1.2 μl) of carbon-coated copper grids (Cu 200 mesh Quantifoil R2/2 or Lacey, EMS), previously glow discharged 2 mA and 1.5–1.8 × 10−1 mbar for 1 min in a glow discharge system (ELMO, Corduan). The sample was then vitrified in a Leica

EMGP system. Briefly, excess liquid was removed by blotting for 6 s at 18 °C and 95% humidity, then the sample was plunge-frozen in liquid ethane. Grids were stored in liquid nitrogen until ready for image acquisition, then loaded onto the 12-grid Autoloader system whilst still under liquid nitrogen.

Dose-symmetrical tilt series were collected on a 200 kV Glacios transmission electron microscope (Thermo Fisher Scientific) equipped with a K2-Summit direct electron detector (Gatan). Tilt series with an angular increment of 2° and an angular range of ±60° were acquired with the tomography software (Thermo Fisher Scientific), v.5.2.0.5806REL. The total electron dose was 130 electrons per Å2 at a pixel size of 8 Å (0.8 nm). Dose-symmetrical tilt series were saved as separate stacks of frames, then aligned using the gold bead markers, motion corrected, and re-stacked from −60° to +60° using IMOD's function alignframes[53]. The 3D reconstructions were calculated in IMOD 4.9.10 by weighted back- projection and a Gaussian filter was used to enhance contrast.

### Structural modelling

For the prediction of protein domains, SMART (Simple Modular Architecture Research Tool)[54] and I-TASSER[55] were used. AlphaFold2[29] was used to generate structural predictions of the Mla proteins in *V. parvula*. To specifically analyse the presence of β-barrels within our target sequences, the BOCTOPUS2 database was used[36]. As another transmembrane site–determining web-based tool, TMHMM 2.0 (Transmembrane Hidden Markov Model) was also used to analyse the transmembrane sites of proteins[56].

All structure predictions with AlphaFold2 were performed through the ColabFold[57] (Colabfold v1.5.2) implementation. Output models of monomeric full-length MlaD (5 models) were ranked according to the pLDDT score. MlaEFD$_{1-130}$ predictions (5 models) were obtained without structural templates using AlphaFold2-multimer v2.3.1 and sorted by *multimer* score[30]. To impose the 2:2:6 stoichiometry of the complex, 2 copies of the MlaE and MlaF sequences were concatenated together with 6 copies of the MlaD$_{1-130}$ sequence. The AlphaFold model of MlaEFD$_{1-130}$ can be found in the Supplementary Dataset 2. Model of the hexameric full-length MlaD complex was obtained by stitching together predictions for the MlaD$_{1-130}$, MlaD$_{36-263}$ and MlaD$_{239-419}$ domains, via structural superimposition of the overlapping regions. Predictions for the MlaD$_{36-263}$ and MlaD$_{239-419}$ domains were first obtained using the AlphaFold2-multimer v2.3 parameters with templates found in the PDB70 database[58] and ranked by *multimer* score[30]. The AlphaFold model of hexameric MlaD can be found in the Supplementary Dataset 3.

Full-length monomeric models of long MlaD sequences were obtained with AlphaFold2 as described above. Secondary structure class of all positions in the best model (highest plDDT) for each sequence were obtained using DSSP[59] and PROSS[60]. To automatically assess the presence of residues in β-strand conformation after the MCE domain, the start and end positions of the MCE domain in each sequence was obtained using HMMSEARCH[61] (HMMSEARCHv3.3.1) with the "MlaD protein" pfam profile (PF02470).

### Phylogenetic analysis

We assembled a databank of 1,083 genomes representing all bacterial phyla present at the National Center for Biotechnology (NCBI April 2020)[16]. We selected three species per order for each phylum. The number of genomes per phylum therefore reflects their taxonomical diversity (Supplementary Dataset 1). We chose preferably genomes from reference species and the most complete assemblies. We then queried this databank for the presence of MlaA, MlaB, MlaC, MlaD, MlaE, MlaF, and TamB, using the pfam domains PF04333, PF05494, PF02470, PF02405 and PF04357 respectively. The alignments of protein families NF033618 (MlaB) and PRK11831 (MlaF) were downloaded from NCBI (Conserved Domain Database) and used to build HMM profiles using HMMBUILD from the HMMERv3.3.2 package[61]. MlaA,

MlaC, MlaD, MlaE and TamB homologs were searched using HMMSEARCH from the HMMERv3.3.2 package. As MlaB and MlaF belong to large protein families, we used MacSyFinder2[32] to identify their occurrences when they are in genomic synteny with at least one of the other components (*mlaACDE*) of the Mla system, with no more than five genes separating them. This also allowed the identification of the occurrences of *mlaACDE* when they were in a genomic cluster. All the retrieved hits were curated using functional annotations, domains organisation, alignments, and phylogeny. We identified 1119 MCE-domain-containing sequences (Supplementary Dataset 1). Among these, homologues of PqiB and LetB (containing more than one MCE domain) were identified only within Proteobacteria and closely related phyla. (Supplementary Fig. 13).

For each protein, curated sequences were aligned with MAFFT v7.407[62] with the L-INS-I option and trimmed using BMGE-1.12[63] with the BLOSUM30 substitution matrix. Maximum likelihood trees were generated for each protein using IQ-TREE.2.0.6[64] with the best evolutionary model assessed with ModelFinder[65] according to the Bayesian Information Criterion, and ultrafast bootstrap supports computed on 1000 replicates of the original dataset. Finally, the presence or absence of each protein was mapped onto a reference tree of Bacteria taken from[16]. All tree figures were generated using custom-made scripts and iTOL[66] (iTOLv6.8.1).

### Statistics and reproducibility

Data are presented as the mean ± standard deviation (SD) and with individual data points from at least three independent biological experiments. Statistical analyses were performed using Prism 9.5.0 (GraphPad Software Inc.). The distribution of data was analysed by Shapiro-Wilk test and their variance using Bartlett test. Since some of the data generated did not follow a normal distribution or had two high variance we performed nonparametric two-tailed Mann-Whitney test. $P$ values < 0.05 were defined as the level of statistical significance.

### Reporting summary

Further information on research design is available in the Nature Portfolio Reporting Summary linked to this article.

## Data availability

The authors declare that all data supporting the findings of this study are available within the paper and its supplementary information files. Phylogenomic analysis data are accessible on Mendeley: https://doi.org/10.17632/pj7gmk86cf.1. Source data are provided with this paper.

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

## Acknowledgements

We gratefully thank Noa Guzzi, and members of the UGB and EBMC units, for their assistance in generating the transposon mutant libraries used in this study. We also thank Robert Smith for his help in scaling up large anaerobic cultures for *V. parvula*. We thank Anna Sartori-Rupp, Stéphane Tachon and Jerzy Witwinowski for help with cryo-EM image analysis. We thank Pierre-Henri Commere for help with the use of the NanoFCM machine. We also thank Philippe Delepelaire for generously providing us with anti-SecA and anti-TolC serum, and Jean-Michel Betton for help in developing the membrane fractionation protocol. This work was supported by funding from the French National Research Agency (ANR) (Fir-OM ANR-16-CE12-0010) and (OM-LipAsy-CE44-008) to CB and SG, by the French government's Investissement d'Avenir Program, Laboratoire d'Excellence "Integrative Biology of Emerging

Infectious Diseases" (grant n°ANR-10-LABX-62-IBEID) and by the Fondation pour la Recherche Médicale (grant DEQ20180339185) to JMG, and by the IR INFRANALYTICS FR2054 to YR and YG. KG was supported by a PhD fellowship of the Pasteur Paris University program (PPU). B-Benyahia was supported by a MENESR (Ministère Français de l'Education Nationale, de l'Enseignement Supérieur et de la Recherche) PhD fellowship. The authors acknowledge the IT department at Institut Pasteur, Paris, for providing computational and storage services (TARS cluster).

## Author contributions

K.G. performed all molecular biology and microbiology experiments with the assistance of B.A. B-Benyahia developed and optimised protocols in *Veillonella parvula* for membrane fractionation and lipid extraction together with K.G. N.T. performed the evolutionary and phylogenetic analyses. B-Bardiaux performed the structural modelling analysis. Y.R., X.T., Z.C. and Y.G. performed lipidomic analyses. J.M.G. provided lab facilities. N.I.P. provided lab facilities and expertise for protein purification, assisted by M.L. C.B. and S.G. supervised the study. K.G., C.B., and S.G. wrote the paper with contributions from N.T., B-Bardiaux and J.M.G. All authors contributed to the final version of the manuscript.

## Competing interests

The authors declare no competing interests.

## Additional information

[1]Institut Pasteur, Université Paris Cité, Genetics of Biofilms Laboratory, Paris, France. [2]Institut Pasteur, Université Paris Cité, Evolutionary Biology of the Microbial Cell Laboratory, Paris, France. [3]Institut Pasteur, Université Paris Cité, Bioinformatics and Biostatistics Hub, F-75015 Paris, France. [4]Institut Pasteur, Université Paris Cité, Structural Bioinformatics Unit, CNRS UMR 3528, Paris, France. [5]Institut Pasteur, Université Paris Cité, Bacterial Transmembrane Systems Unit, CNRS UMR 3528, Paris, France. [6]Université de Lille, CNRS, UMR 8576 – UGSF – Unité de Glycobiologie Structurale et Fonctionnelle, Lille, France. [7]Université de Lille, CNRS, INRAE, Centrale Lille, Université d'Artois, FR 2638 – IMEC – Institut Michel-Eugène Chevreul, Lille 59000, France. [8]Institute for Glyco-core Research (iGCORE), Gifu University, Gifu, Japan. [9]These authors contributed equally: Basile Beaud Benyahia, Najwa Taib, Bianca Audrain, Benjamin Bardiaux. ✉e-mail: simonetta.gribaldo@pasteur.fr; christophe.beloin@pasteur.fr

