## [Peer Review File · Nature Communications]

The Mla system of diderm Firmicute Veillonella parvula reveals an ancestral transenvelope bridge for phospholipid traffickingReviewer #1 (Remarks to the Author):

In this work "Bridges instead of boats? The Mla system of diderm Firmicute *Veillonella parvula* reveals an ancestral transenvelope core of phospholipid trafficking.", Grasekamp, et al. investigate inter membrane phospholipid trafficking in the diderm Firmicute, *Veillonella parvula*. The authors identify a system with homology to part of the *E. coli* Mla system and determine that this system functions in retrograde phospholipid transport using a somewhat different mechanism than *E. coli*, mediated by a trans envelope architecture of MlaD. Further, the authors identify suppressor mutants in TamB suggesting that at least some anterograde transport is mediated by TamB.

This paper, which is well-written, should be of interest to a very broad audience as it brings together biochemical, genetic, computational, and evolutionary approaches to investigate intermembrane phospholipid trafficking in a non-model organism and to investigate the application of the finding across bacteria. The findings are significant and the approaches are rigorous. However, there are several points that need clarification.

Comments

1. Ln 107-8: I don't think that it is accurate to claim there is a dogma that all OM biogenesis systems are the same as *E. coli* simple because other species have not been studied. This statement should be softened.
2. The figures numbers are mislabeled in the figure legends.
3. It is mentioned that an *miaE* mutant was not obtained, but a triple mutant was produced. How often does mutant production fail in these species? Is it possible that deletion of *miaE* in the presence the MlaDF is toxic?
4. Ln 143: Was a homolog of *pgsA* identified?
5. Ln 145-6: This should be reworded to avoid confusion over whether all the phospholipid synthesis genes are encoded as part of a fused gene rather than just the plasmalogen genes.
6. Fig. S3A: Reference lipids are listed as PC, PE, and CL in one TLC and the legend and as PG, PE, and CL in the other TLC. Is this accurate? Also, the text mentions whole cell and membrane extracts. It is unclear which are shown.
7. Ln 196-8: Although I agree with the conclusions of the Mla system being involved in retrograde transport and TamB in anterograde transport, there must be at least one more anterograde phospholipid transporter if *tamB* can be deleted (unless the OM is not essential). Are there other AsmA-like proteins in *V. parvula*?
8. Fig. Ext 4A: It would be useful to have a view of the barrel from the top or bottom to be able to see whether the barrel has an open channel and how large the channel is.
9. Ln 281-3: This sentence seems to imply that *M. tuberculosis* is monoderm. It is not gram-negative but does have twofold membranes (cytoplasmic membrane and mycomembrane).
10. Fig. Ext 9: Are the disordered regions of the example MlaD homologs without barrels predicted to form a structure if modeled as a hexamer?

Reviewer #2 (Remarks to the Author):

This study by Grasekamp et al. focuses on understanding the Mla system in *Veillonella*. The Mla system has been mainly studied in the "model" organism *Escherichia coli*. Mla is important for maintaining the asymmetric lipid structure of the outer membrane (OM) by transporting phospholipids from the outer leaflet of the OM to the inner membrane (IM) using several proteins including a soluble periplasmic shuttle protein, MlaC. Interestingly, *Veillonella* lacks MlaC, as well as the OM component MlaA, only sharing the core components of the IM Mla complex. The work presented here clearly shows that the Mla system in *Veillonella* functions like that of *E. coli*; however, it has a different structure. The authors show that one of the IM components, MlaD, is significantly larger in *Veillonella* than in *E. coli*, with a predicted long helical domain and C-terminal b-barrel. By generating AlphaFold and AlphaFold Multimer models and by monitoring protein localization in cells, the authors show that the large *Veillonella* MlaD bridges the IM and OM.

Specifically, their structural models predict that an MlaD hexamer forms a tunnel-like helical structure that could cross the periplasm and connects to six beta-barrels predicted to be in the OM. The authors present phylogenetic studies and propose the evolution of the Mla system. They also point out that the "model" system from *E. coli* is more an exception than the rule.

I enjoyed reading and thinking about this study. The experiments are well done, the data are clear, and the conclusions are justified. The manuscript is also well crafted. Overall, the work is impactful. It clearly demonstrates the great value of studying the cell envelope of "non-model" organisms, especially those that are distantly related to the traditional model organisms such as *E. coli*. Nice work!

There are two main points for authors to consider:

1) What is the estimated length of the predicted tube-like structure that is proposed to cross the periplasm? Is the estimated length in agreement with the size of the periplasm in *Veillonella* cells? This information should be added to the manuscript.

2) OM vesicles: It would be important to include a control that rules out lysis, such as an immunoblot for a cytoplasmic protein. Also, how were samples standardized for the TLC shown in Fig. 1C?

Minor points:

3) Why is the *miaE* single mutant not viable but the tripe *miaEFD* viable? I assume that having MlaD and/or the ATPase MlaF without MlaE is lethal? It will be speculative (and should be stated as such), but it would be nice to include a possible explanation so that readers are not confused.

4) Lines 193-195: "The contrasting OM permeability phenotypes of $\Delta mlaD$ and $\Delta tamB$, together with the striking complementation of the double $\Delta mlaD\Delta tamB$ strain..." I think the use of "complementation" is incorrectly used here because deletions cannot complement anything. Instead, "suppression" or "cosuppression observed in the double mutant" should be used.

5) Lines 202-203: "...understand how three IM proteins could be structurally arranged within the envelope of *V. parvula* to facilitate GPL trafficking in the absence of MlaABC." MlaA is a cytoplasmic protein that forms a complex with two IM proteins. In addition, the question does not really pertain to MlaB (cytoplasmic protein of unclear function). What is unclear is how the Mla system functions without the periplasmic shuttle/chaperone and OM component known to exist in *E. coli*. I find the text confusing. I suggest that the authors change it to something like "...understand how the predicted structure of the IM MlaEFD complex could facilitate GPL trafficking in *V. parvula* in the absence of the periplasmic and OM MlaCA components."

6) Lines 220-221: "generating a model for the full length protein (Fig 4B)." Using what to generate the model? Information about using AlphaFold should be included here and in the legends of Ext. Fig 3 and 4. Also, it should be "full-length protein".

7) Lines 375-378: Reference 40 only showed the structure of a section of TamB. The authors might want to consider referencing recent publications that include AlphaFold predicted structures for TamB and other AsmA-like proteins. For example: PMIDs 37455811, 36571082, and 34781743.

8) The authors should clearly state that the structures presented are models or predicted structures throughout the manuscript. I recommend adding "predicted" or "model" in front of "structure(s)" when referring to models. This is especially important in the Discussion when discussing the MlaD proteins that are large but not predicted to have a beta-barrel. I doubt that the C-termini of those proteins look like spaghetti. Either the model is wrong, or a partner(s) is missing that completes the fold.

9) Check labels in Fig. S3 for i and ii panels. Labels have been cut off or are missing.

Reviewer #3 (Remarks to the Author):

Grasekamp, et al. investigate the possible role of an MCE transport system in the transport of glycerophospholipid (GPL) for the maintenance of the outer membrane in the non-model diderm species, *Veillonella parvula*. Based upon distant similarity to some of the components of the Mla system in *E. coli*, the authors refer to the *V. parvula* system as "Mla" and hypothesize that it may have a similar function in maintaining the asymmetry of the outer membrane; indeed, the authors show that mutations in the *V. parvula* system result in similar phenotypes, such as detergent hypersensitivity, increased vancomycin resistance, and hypervesiculation. The authors also show that loss of tamB in a mlaD KO background partially rescues the detergent hypersensitivity phenotype, suggesting that MlaEFD and TamB play opposing roles (retrograde and anterograde trafficking, respectively) in maintaining GPL homeostasis of the *V. parvula* outer membrane (OM). However, there are substantial differences between *E. coli* Mla and the *V. parvula* system, which makes this work quite interesting. Structural modeling and subcellular localisation experiments suggest that the *V. parvula* MCE protein may form a transenvelope tunnel that is anchored in both the IM (inner membrane) and OM. Intriguingly, the authors use bioinformatics to show that MCE proteins like the one from *V. parvula* are widespread across diderm bacteria, and may perhaps be similar to the most ancestral form of the transporter.

Overall, we feel like this is a very interesting paper that adds to our understanding of this diverse family of transporters. The results are generally clear, experiments appear to be well executed, and the use of AlphaFold was appropriate, thoughtfully analyzed, and presented with the right degree of "skepticism" (it is super powerful, but sometimes does funny things, like it did here in the helical tunnel vs groove predictions). The paper was posted as a pre-print on BioRxiv, is well-written, and was a pleasure to read. We don't have any very serious concerns, but have a number of suggestions for improving the final version.

MAJOR COMMENTS:

None

MINOR COMMENTS:

We would suggest referring to the proposed complex as forming a "tunnel" instead of a "bridge", since bridges usually leave the cargo open to the environment (like the LPS exporter bridge), while tunnels completely surround the cargo (like in an efflux pump).

We think the authors should call the *V. parvula* system something other than "Mla". There is certainly some similarity to *E. coli* Mla, but there are also substantial differences (as the authors point out, only 3 of the 7 proteins found in *E. coli* Mla have counterparts in the *V. parvula* system). It is also worth noting that there are bacterial species like *P. aeruginosa* that have a bonafide Mla system as well as a *V. parvula*-like MCE system, which would make calling both "Mla" a bit awkward. I think the best characterized *V. parvula*-like MCE system is called TGD in *A. thaliana*; perhaps a new name could be based on that, or could be something completely new of the authors choosing. I think this will help avoid confusion in the field.

The authors show that the *V. parvula* mla KO strains exhibit increased resistance to vancomycin, but this phenotype does not appear to be complemented by adding back in the deleted genes. Was a similar pattern observed for *E. coli* mla mutants? Why can't this be complemented?

SDS/EDTA sensitivity assay: The concentrations of SDS reported (~0.003-0.004%) are ~100-fold lower than what is typically used for *E. coli* mla mutants. We would like to confirm that these values are correct and not a typo.

Have the authors tried predicting a homohexamer of the beta-barrel domain? If they are

analogous to MlaA, they would need to create a pathway for lipids to move from within the OM to the tunnel through the MCE protein. It would be interesting if AF suggests they form a stable assembly, vs each barrel doing its own thing in isolation.

In 71: Perhaps it would be appropriate to ref the earlier work establishing the IM complex here: Kamischke 2019, Ekiert 2017, and Thong 2016; if there is space, one could also cite the more recent 6 papers for the high resolution structures that all came out around the same time (PMIDs: 33845086, 34188171, 33298869, 33199922, 32884137, 33236984). In the context of the MlaA-Omp complex, we could also imagine citing Abellon-Ruiz 2017 here.

Fig. 1 legend: The authors refer to the MCE protein as a "substrate binding protein", akin to MBP in a classical ABC importer. However, the MCE protein is playing a completely different role (if anything, MlaC would be analogous to a SBP, but that is absent in the *V. parvula* system). We suggest removing this terminology.

Line 118-121: The authors mention making single mlaDEF mutations and that all *V. parvula* mla mutants display hypersensitivity to SDS/EDTA and increased resistance to vancomycin. However, not all of this data is shown, so this claim is not well supported. We suggest showing all the data or altering the claim to match the data shown.

In 131: "these phenotypes are indicative of a loss of lipid asymmetry": We might suggest softening the claim here. This may be correct, but OM asymmetry has not been directly assessed, and this system appears to be quite different from mla. It may be remodeling the OM in a different way, and we think the observed phenotypes don't necessarily have to arise from a loss of asymmetry.

In 151-153: Is there any precedent for vinyl ether bonds in bacterial lipids? Perhaps a reference to prior work here would be appropriate, and a statement of how common/uncommon these are?

In 176-177: Are these 2 independent insertions at the exact same location?

In 179-180: How many AsmA-like proteins are encoded in the genome? This may affect the claim made in In 196-198, which could perhaps be softened a bit to avoid being overly speculative.

In 203-204: Does that mean that MlaF has a long C-terminal tail that may "handshake" with the neighboring MlaF subunit, even in the absence of MlaB? Or is the *V. parvula* MlaF a little shorter? Also, in other MlaE structures there is an N-terminal amphipathic helix that runs parallel to the plane of the membrane, in the cytoplasmic leaflet. I can see it in Ext. Data Fig. 3D for *E. coli* but can't pick it out for the *V. parvula* prediction. Is it missing in *V. parvula* MlaE? I think it would be worth 1 sentence commenting on these differences if they are indeed different.

In 281-283: We think mycobacteria are regarded by many in the field as diderm (though I have had heated debates about this with colleagues), while we agree that many other actinobacteria are monoderms. Cryo ET of mycobacteria reveals an OM that appears very much like the OM of Gram-negatives (e.g., Hoffman, PNAS, 2008). So we suggest removing the specific mention of *M. tuberculosis* as a monoderm.

Main Text Figures: The middle 3 Figures need to be re-labeled to match the in-text references (the second Fig. 1 → Fig. 2, the current Fig. 2 → Fig. 3 and the current Fig. 3 → Fig. 4).

Fig 3A: Are the strips for each strain from the same plate, or is this a composite image? In order to compare the suppressor mutants to the controls, they need to be on the same plate, but I'm not sure that this is the case?

Fig. 3C: Please indicate in the legend what the dashed line box is highlighting (i.e. the hydrophobic regions that are predicted to be membrane-embedded). Also, perhaps this is due to the angle of the model, are there only three C-terminal beta barrels (reflecting the possibility that several beta-barrels can fuse to form a trimeric conformation, as mentioned in the Discussion) or are another three just hidden from view?

Fig. 5: We think it would be very helpful to have three different categories of "MlaD" proteins, and color code them differently (currently, they are all one shade): 1) true MlaD proteins (with short C-terminal helical region); 2) Single MCE domain proteins with a long helical region, with beta-barrels; and 3) Single MCE domain proteins with a long helical region, WITHOUT beta-barrels.

Extended Fig. 3A: Label the figure with "i" and "ii" as written in the legend. Also, the legend mentions that the major lipid species in *V. parvula* are labeled 1-4 in the figure, however the #4 in Aii looks like it runs similarly to the cardiolipin standard. This is a little confusing as the text mentions that there is no cardiolipin detected by MALDI-QIF-TOF.

Extended Fig. 3Bi: The sentence "The relative decrease in these two species from the mla KO OMVs confirmed the relative enrichment of PE" is a little confusing. Is there a typo here or can the authors elaborate more on the connection between the reduction of unknown lipid and PE increase.

Extended Fig. 5: In the legend, A, B, D, E refer to hexameric chains A to G. Should this not be A to F, as labeled in the figure? It would be more useful in understanding the figure if the legend detailed what each dashed line box was highlighting (around rank 4 of A, B, C and the AlphaFold models of C and F). Also, having the PAE colored error bar labeled with "high error" and "low error" as in Ext. Fig. 3C would be helpful (can only label one representative error bar so as to not clutter the figure).

Extended Fig. 6: "Presence OF absence of Mla components..." I believe should be "Presence OR absence". We also cannot read the detailed species names, as well as for Ext. Fig. 7.

In order to make the AlphaFold predictions readily accessible to the community, coordinates should either be deposited in ModelArchive or included as supplementary files as part of the publication.

REBUTTAL LETTER

REVIEWER COMMENTS

Reviewer #1 (Remarks to the Author):

In this work “Bridges instead of boats? The Mla system of diderm Firmicute *Veillonella parvula* reveals an ancestral transenvelope core of phospholipid trafficking.”), Grasekamp, et al. investigate inter membrane phospholipid trafficking in the diderm Firmicute, *Veillonella parvula*. The authors identify a system with homology to part of the *E. coli* Mla system and determine that this system functions in retrograde phospholipid transport using a somewhat different mechanism than *E. coli*, mediated by a trans envelope architecture of MlaD. Further, the authors identify suppressor mutants in TamB suggesting that at least some anterograde transport is mediated by TamB.

This paper, which is well-written, should be of interest to a very broad audience as it brings together biochemical, genetic, computational, and evolutionary approaches to investigate intermembrane phospholipid trafficking in a non-model organism and to investigate the application of the finding across bacteria. The findings are significant and the approaches are rigorous. However, there are several points that need clarification.

Comments

1. Ln 107-8: I don't think that it is accurate to claim there is a dogma that all OM biogenesis systems are the same as *E. coli* simple because other species have not been studied. This statement should be softened.

Answer:

We have softened this statement by replacing this sentence by: “Together, our results uncover novel functional information about GPL trafficking in a non-model organism, shedding light on the evolution of the Mla system, challenging the assumption that OM biogenesis systems studied in *E. coli* represent the majority of diderm bacteria, and highlighting the diversity in the OM biogenesis and maintenance systems.”

2. The figures numbers are mislabeled in the figure legends.

Answer:

Thank you, this has been corrected.

3. It is mentioned that an *miaE* mutant was not obtained, but a triple mutant was produced. How often does mutant production fail in these species? Is it possible that deletion of *miaE* in the presence the MlaDF is toxic?

Answer:

Although *V. parvula* is not always the easiest microorganism to manipulate, construction of mutants rarely fails in this species unless we encounter a problem of toxicity or essentiality. We favour the same hypothesis as one of the other reviewers to explain our incapacity to construct the *miaE* mutant, and have added this hypothesis at lines 113-114:

“— with the exception of the $\Delta miaE$ mutant that could not be obtained, perhaps due to a toxicity of MlaD and/or MlaF in the absence of MlaE —“.

4. Ln 143: Was a homolog of *pgsA* identified?

Answer:

As described in supplementary Fig. 1A (now supplementary Fig. 3), several homologues of genes involved in glycerophospholipid biosynthesis have been identified in *V. parvula*. FNLLGLLA_00889 encodes a protein that displays homology to both Pss and PgsA, though further work is required to determine its function. This has been corrected in the now Supplementary Fig. 3A.

5. Ln 145-6: This should be reworded to avoid confusion over whether all the phospholipid synthesis genes are encoded as part of a fused gene rather than just the plasmalogen genes.

Answer:

Thank you - we have split the sentence to avoid confusion. Only the plasmalogen gene appears to encode the function of both PlsA and PlsR.

6. Fig. S3A: Reference lipids are listed as PC, PE, and CL in one TLC and the legend and as PG, PE, and CL in the other TLC. Is this accurate?

Answer:

Thank you for pointing out this typing mistake. This has been corrected (PC for both TLCs).

Also, the text mentions whole cell and membrane extracts. It is unclear which are shown.

Answer:

Thank you, only whole-cell extracts are shown for the first panel (A) and the secondary panels show OMV-derived extracts – we have now added details of this in the figure legend and title in Supp Fig 3 (now renamed Supp Fig 5).

7. Ln 196-8: Although I agree with the conclusions of the Mla system being involved in retrograde transport and TamB in anterograde transport, there must be at least one more anterograde phospholipid transporter if tamB can be deleted (unless the OM is not essential). Are there other AsmA-like proteins in *V. parvula*?

Answer:

Yes, we did identify one other gene encoding a potential AsmA-like protein in *Vp*. We constructed all combinations of mutants of *mldD*, *tamB* and “*asmaA*”. *AsmA* does not appear to have a significant impact on OM homeostasis in the combinations of deletion that were tested. Indeed, even the triple $\Delta mldD \Delta tamB \Delta asmaA$

mutant is viable, suggesting there are other unidentified PL trafficking systems in *V. parvula*. We also see that deletion of *asmA* in the *mldA* background does not suppress the associated OM permeability phenotype, nor that of the hypervesiculation.

Considering that this gene was also not identified in any suppressor or transposon mutagenesis screens for the *mldA* mutant, we decided not to introduce supplementary information on it to simplify our message. However, we will likely use this information for a future study as we continue to probe the systems which govern PL homeostasis in *V. parvula*.

We have added the information of the existence this AsmA-like protein in Vp at the end of the discussion section and indicate that further work will be necessary to clarify which function are exactly performed by TamB and this AsmA-like protein and to identify potential other genes involved in PLs transport in Vp:

Lines 363-363: "Indeed, one other *asmA*-like protein exists in *V. parvula*, and further work is required to elucidate its functional role, if any, in the context of OM homeostasis."

8. Fig. Ext 4A: It would be useful to have a view of the barrel from the top or bottom to be able to see whether the barrel has an open channel and how large the channel is.

Answer:

As the pore channel is quite narrow and not straight, a top or bottom view does not allow to see the channel. Instead, we now present in Supplementary Fig. 9C (formerly Extended Fig4C) a representation of the channel as computed by the HOLE software.

9. Ln 281-3: This sentence seems to imply that *M. tuberculosis* is monoderm. It is not gram-negative but does have twofold membranes (cytoplasmic membrane and mycomembrane).

Answer:

Thank you for this comment. Yes the mycomembrane is clearly an outer membrane, but emerged "de novo" in the clade to which *M. tuberculosis* belongs. We were referring to the phylum as being ancestrally monoderm. We agree that this can be confusing and have removed the sentence for simplicity.

10. Fig. Ext 9: Are the disordered regions of the example MlaD homologs without barrels predicted to form a structure if modeled as a hexamer?

Answer:

Thank you for this interesting remark.

Using AlphaFold2-Multimer, we predicted the structure of hexamers for the C-terminal domain (including the last segment of the helical stalk) of the 2 examples MlaD homologues shown in Extended Fig 9 (now Supplementary Fig. 14) (526226_Gbronchialis@ACY20515.1 and 83332_Mtuberculosis@CCP43333.1). In both cases, the confidence scores returned by AlphaFold (pLDDT, pTM and ipTM) were very low and no additional folding was observed. Of note, 83332_Mtuberculosis@CCP43333.1 has 65% identity with *M. smegmatis* mce1f protein from the Mce1 transporter for which the structure was reported recently (Chen et al., Nature. 2023 Aug;620(7973):445-452. doi: 10.1038/s41586-023-06366-0) and in which the C-ter of the 6 different Mce1 proteins form a small barrel. We thus cannot exclude that some of the "without barrel" MlaD homologues analysed here do not self-associate to form a hexamer but rather form a complex with other MlaD-related proteins. However, our AlphaFold modelling as hexamer of 16 more "without barrel" MlaD sequences revealed that two of them possibly form tiny barrels, with a hydrophobic core. This further suggests a potential convergent evolution leading to homo- or heteromeric β -barrels.

This information is now provided in a new Supplementary Figure 15 (see below) and further discussed in lines 307-311.

“Interestingly, many of the long MlaD sequences with no identifiable complete β -barrel domain contain a predicted disordered C-terminal region with some β -structure, often short β -hairpins, which may represent remnants or precursors of a β -barrel (**Supplementary Fig 14**). For two of these barrel-less sequences, homo-hexameric predictions generated by AlphaFold revealed a tiny 6-stranded C-terminal barrel formed by each monomer contributing a single β -strand (**Supplementary Fig 15**). A similar conformation was recently structurally characterised in *M. smegmatis*³¹, suggesting barrels may be formed through multimerisation, even in some of the seemingly barrel-less MlaD sequences.”

Reviewer #2 (Remarks to the Author):

This study by Grasekamp et al. focuses on understanding the Mla system in *Veillonella*. The Mla system has been mainly studied in the “model” organism *Escherichia coli*. Mla is important for maintaining the asymmetric lipid structure of the outer membrane (OM) by transporting phospholipids from the outer leaflet of the OM to the inner membrane (IM) using several proteins including a soluble periplasmic shuttle protein, MlaC. Interestingly, *Veillonella* lacks MlaC, as well as the OM component MlaA, only sharing the core components of the IM Mla complex. The work presented here clearly shows that the Mla system in *Veillonella* functions like that of *E. coli*; however, it has a different structure. The authors show that one of the IM components, MlaD, is significantly larger in *Veillonella* than in *E. coli*, with a predicted long helical domain and C-terminal β -barrel. By generating AlphaFold and AlphaFold Multimer models and by monitoring protein localization in cells, the authors show that the large *Veillonella* MlaD bridges the IM and OM. Specifically, their structural models predict that an MlaD hexamer forms a tunnel-like helical structure that could cross the periplasm and connects to six β -barrels predicted to be in the OM. The authors present phylogenetic studies and propose the evolution of the Mla system. They also point out that the “model” system from *E. coli* is more an exception than the rule.

I enjoyed reading and thinking about this study. The experiments are well done, the data are clear, and the conclusions are justified. The manuscript is also well crafted. Overall, the work is impactful. It clearly demonstrates the great value of studying the cell envelope of “non-model” organisms, especially those that are distantly related to the traditional model organisms such as *E. coli*. Nice work!

There are two main points for authors to consider:

1) What is the estimated length of the predicted tube-like structure that is proposed to cross the periplasm? Is the estimated length in agreement with the size of the periplasm in *Veillonella* cells? This information should be added to the manuscript.

Answer:

From the top of the TM-helix to the bottom of the OM beta-barrel ring, the length is 18.5 nm, slightly lower than the commonly reported periplasmic width of 21 nm in *E coli* doi.org/10.1128/jb.185.20.6112-6118.2003). However, the beta-hairpin connecting the last central helix and the beta-barrel (displaying low pI DDT values) could adopt a more extended conformation, adding up 5 nm more, for a total possible length of 23.5 nm. An accurate measurement of Vp periplasm size using cryo-EM images leads to a width of 24.2 ± 1.8 nm (n=26), so the maximum length of the tube-like structure is compatible with our estimation of the Vp periplasm width.

This has been added to the manuscript (lines 231-233):

“Like these resolved structures, the predicted tunnel formed by the α -helices generates a hydrophobic interior of ~ 13 Å in diameter (Fig 4C) - supporting the possibility of a role in hydrophobic substrate transport - and a total possible length of ~ 23.5 nm, compatible with the periplasmic width in *V. parvula* (24.2 ± 1.8 nm (n=26) as measured by cryo-EM.”

2) OM vesicles: It would be important to include a control that rules out lysis, such as an immunoblot for a cytoplasmic protein. Also, how were samples standardized for the TLC shown in Fig. 1C?

Answer:

Thank you for highlighting this aspect. As evidence that rules out lysis, we see none of the Δmla mutants display increased lysis via efficiency of plating (cfu counting) (see for example Fig 2, Supplementary Figs 2 & 6), which is also supported by little difference in RNAseq profiles between WT and Δmla strains, even under conditions of slight stress (data not shown). More importantly, however, our NanoFCM data generated from both whole-cell cultures, and of extracted and purified OMV samples, display the same average scattering profiles for ‘small events’ or vesicles, with an average diameter of ~ 60 nm (as shown in Supplementary Fig 7), suggesting that these extracted samples are indeed vesicles and not random fragments of cell membrane that recombine. This is further supported by the average 60 nm diameter of OMVs observed in tomograms generated from $\Delta mlaD$ cultures.

This information is now provided in the Methods section:

Lines 615-616: “Purified OMVs were analysed via NanoFCM confirming an average size of ~ 60 nm, matching the average size of vesicles in whole cell cultures observed via both NanoFCM and cryo-electron tomography (Supplementary Fig 7). These observations of OMVs preparation are compatible with absence of cell lysis as also observed in cfu counting experiments of WT and $\Delta mlaD$ strains (see Fig 2, Supplementary Figs 2 & 6).”

To rule out the presence of lysis, we did indeed attempt immunoblotting against OMV samples with antibodies against both TolC and SecA, as previously used for the membrane fraction localisation. Expectedly, no signal was generated for SecA, the inner membrane control. However, no signal was found for TolC, the OM control, either. We posit that TolC may be excluded from vesicle formation as it is connected to inner membrane proteins. This would not prevent it from being used as an OM marker for fractionation because of the French press lysis process, but makes it a very inconvenient marker for OMVs. We are trying to obtain a new antibody targeting the OM for our OmpM related studies, which may yield better results, but this is out of scope for this paper. Another ongoing development is the use of the enzymatic activity of catalase and NADH oxidase as a reporter for the presence of cytoplasmic and periplasmic membrane contamination in OMVs or in membrane fractionation assays.

For the standardisation of TLCs, lipids were extracted from the same volume of cell cultures normalised by OD600 (or in the case of OMVs, the same volume of supernatant) and resultant extracted lipid pellets were weighed and resuspended in the corresponding volume of chloroform before loading the same volume of each extracted lipid sample (10 μ l) onto the silica plates. However, taking into account both the variabilities in lipid extraction and weighing small quantities of lipids, we think it is more cautious to refer to the relative enrichment of PE within a sample as compared to other lipid species - such as the ratio of PE to lipid 2 - rather than quantities.

We have added the information in the Methods section and rephrased the manuscript accordingly:

Lines 158-160: "Across all iodine vapour-stained plates, OMVs produced by $\Delta mlaD$ strains display a relative enrichment for PE when compared to another lipid species that is abundant in WT OMVs (Fig 2c, Supplementary Fig 5b)."

As we continue to develop our toolkit for lipid analyses in *V. parvula*, we will generate more quantitative data to further interrogate the functioning of these genes in OM homeostasis and maintenance.

Minor points:

3) Why is the *miaE* single mutant not viable but the triple *miaEFD* viable? I assume that having MlaD and/or the ATPase MlaF without MlaE is lethal? It will be speculative (and should be stated as such), but it would be nice to include a possible explanation so that readers are not confused.

Answer:

Thank you for pointing to this aspect. As stated in our answer to reviewer 1, we favour the hypothesis that having MlaD and/or the ATPase MlaF without MlaE is toxic for the cell.

We have added this hypothesis in lines 113-114:

"— with the exception of the $\Delta mlaE$ mutant that could not be obtained, perhaps due to a toxicity of MlaD and/or MlaF in the absence of MlaE —«.

4) Lines 193-195: "The contrasting OM permeability phenotypes of $\Delta mlaD$ and $\Delta tamB$, together with the striking complementation of the double $\Delta mlaD\Delta tamB$ strain..." I think the use of "complementation" is incorrectly used here because deletions cannot complement anything. Instead, "suppression" or "cosuppression observed in the double mutant" should be used.

Answer:

Thank you for this comment – we have modified this sentence for clarity:

"The contrasting OM permeability phenotypes of $\Delta mlaD$ and $\Delta tamB$, together with the striking suppression of the $\Delta mlaD$ OM related phenotype by the *tamB* mutation, hints to the antagonistic functions of these two genes in envelope biogenesis and maintenance."

5) Lines 202-203: "...understand how three IM proteins could be structurally arranged within the envelope of *V. parvula* to facilitate GPL trafficking in the absence of MlaABC." MlaA is a cytoplasmic protein that forms a complex with two IM proteins. In addition, the question does not really pertain to MlaB (cytoplasmic protein of unclear function). What is unclear is how the Mla system functions without the periplasmic shuttle/chaperone and OM component known to exist in *E. coli*. I find the text confusing. I suggest that the authors change it to something like "...understand how the predicted structure of the IM MlaEFD complex could facilitate GPL trafficking in *V. parvula* in the absence of the periplasmic and OM MlaCA components."

Answer:

Thank you for this proposition. We have modified the sentence as suggested.

6) Lines 220-221: "generating a model for the full length protein (Fig 4B)." Using what to generate the model? Information about using AlphaFold should be included here and in the legends of Ext. Fig 3 and 4. Also, it should be "full-length protein".

Answer:

We have now modified the sentence from:

"Considering all characterised MCE proteins to date form hexamers, we merged overlapping predicted structures of all domains of MlaD_{vp} with hexameric stoichiometry, generating a model for the full-length protein (Fig 4b)."

To:

“Considering all characterised MCE proteins to date form hexamers, we built a structural model of the full-length protein by merging models obtained with AlphaFold-Multimer^{29,30} of hexameric configuration of all domains of MlaDVp with overlapping segments (Fig 4b).

We also have modified the legend of the corresponding figures accordingly.

7) Lines 375-378: Reference 40 only showed the structure of a section of TamB. The authors might want to consider referencing recent publications that include AlphaFold predicted structures for TamB and other AsmA-like proteins. For example: PMIDs 37455811, 36571082, and 34781743.

Answer:

Thank you - we have now added the corresponding publications.

8) The authors should clearly state that the structures presented are models or predicted structures throughout the manuscript. I recommend adding "predicted" or "model" in front of "structure(s)" when referring to models. This is especially important in the Discussion when discussing the MlaD proteins that are large but not predicted to have a beta-barrel. I doubt that the C-termini of those proteins look like spaghetti. Either the model is wrong, or a partner(s) is missing that completes the fold.

Answer:

Thank you for your comment; we have added "predicted" where necessary. We agree with the reviewer that there is likely much to discover regarding this C-terminal region of long "barrel-less" MlaD, and we elaborated on this already in the discussion section, as well as the possibility of accessory proteins required for the proper folding / functioning of these transenvelope proteins.

9) Check labels in Fig. S3 for i and ii panels. Labels have been cut off or are missing.

Answer:

Thank you, this has been modified.

=====
Reviewer #3 (Remarks to the Author):

Grasekamp, et al. investigate the possible role of an MCE transport system in the transport of glycerophospholipid (GPL) for the maintenance of the outer membrane in the non-model diderm species, *Veillonella parvula*. Based upon distant similarity to some of the components of the Mla system in *E. coli*, the authors refer to the *V. parvula* system as "Mla" and hypothesize that it may have a similar function in maintaining the asymmetry of the outer membrane; indeed, the authors show that mutations in the *V. parvula* system result in similar phenotypes, such as detergent hypersensitivity, increased vancomycin resistance, and hypervesiculation. The authors also show that loss of tamB in a mlaD KO background partially rescues the detergent hypersensitivity phenotype, suggesting that MlaEFD and TamB play opposing roles (retrograde and anterograde trafficking, respectively) in maintaining GPL homeostasis of the *V. parvula* outer membrane (OM). However, there are substantial differences between *E. coli* Mla and the *V. parvula* system, which makes this work quite interesting. Structural modeling and subcellular localisation experiments suggest that the *V. parvula* MCE protein may form a transenvelope tunnel that is anchored in both the IM (inner membrane) and OM. Intriguingly, the authors use bioinformatics to show that MCE proteins like the one from *V. parvula* are widespread across diderm bacteria, and may perhaps be similar to the most ancestral form of the transporter.

Overall, we feel like this is a very interesting paper that adds to our understanding of this diverse family of transporters. The results are generally clear, experiments appear to be well executed, and the use of AlphaFold was appropriate, thoughtfully analyzed, and presented with the right degree of "skepticism" (it is super powerful, but sometimes does funny things, like it did here in the helical tunnel vs groove predictions). The paper was posted as a pre-print on BioRxiv, is well-written, and was a pleasure to read. We don't have any very serious concerns, but have a number of suggestions for improving the final version.

MAJOR COMMENTS:

None

MINOR COMMENTS:

- We would suggest referring to the proposed complex as forming a “tunnel” instead of a “bridge”, since bridges usually leave the cargo open to the environment (like the LPS exporter bridge), while tunnels completely surround the cargo (like in an efflux pump).

Answer:

Thank you for this suggestion, and we agree with the reviewer on the formal definition of bridge vs tunnel. The term bridge that we used in the title, and to describe what we think could be a transenvelope system, is used here more as a generic term. As indicated by the reviewer, predicted structures by AlphaFold are models and we cannot today be sure that the real structure taken by Vp MlaD is actually a closed tunnel. The use of the term tunnel in the title could therefore be misleading, while using a generic term bridge (e.g. a connecting structure between the two membranes) seems more appropriate at this stage of our knowledge.

- We think the authors should call the *V. parvula* system something other than “Mla”. There is certainly some similarity to *E. coli* Mla, but there are also substantial differences (as the authors point out, only 3 of the 7 proteins found in *E. coli* Mla have counterparts in the *V. parvula* system). It is also worth noting that there are bacterial species like *P. aeruginosa* that have a bonafide Mla system as well as a *V. parvula*-like MCE system, which would make calling both “Mla” a bit awkward. I think the best characterized *V. parvula*-like MCE system is called TGD in *A. thaliana*; perhaps a new name could be based on that, or could be something completely new of the authors choosing. I think this will help avoid confusion in the field.

Answer:

Although we understand this comment, we feel renaming these proteins – which are homologous to characterized Mla proteins and still perform Mla (maintenance of lipid asymmetry) functions – would bring about more confusion in the field. Considering MlaEFD are the initially occurring proteins in evolution, and that this system has adapted and become more complex with the addition of MlaABC in the Proteobacteria, we are comfortable in maintaining MlaEFD as descriptors. This is also necessary considering the high homology between MlaE, MlaF, and the first ‘half’ of MlaD, in our new model and in other classical diderms.

- The authors show that the *V. parvula* *mia* KO strains exhibit increased resistance to vancomycin, but this phenotype does not appear to be complemented by adding back in the deleted genes. Was a similar pattern observed for *E. coli* *mia* mutants? Why can't this be complemented?

Answer:

We agree that this lack of complementation is unusual, and do not yet fully understand this phenomenon. Interestingly, vancomycin resistance of *mia*-deficient strains has been described in several other bacterial strains – including *E. coli*, *A. baumannii* and *N. gonorrhoeae* – and is so widely accepted that results published by Kamischke et al. were refuted by reviewers, as the authors initially reported an increased sensitivity of *A. baumannii* Δ *mia* mutants to vancomycin (Kamischke et al., 2019). Further, the increased vancomycin resistance of Δ *miaA* mutants of *N. gonorrhoeae* were also unable to be complemented by reintroducing *miaA* into the mutant (Baarda et al., 2019).

We propose that deeper, global envelope changes are triggered by the deletion of these *mia* genes, possibly involving a complex network of other pathways that cannot be restored upon reintroduction of Mla on a plasmid vector (notably due to problems in restoring the exact stoichiometry of the Mla proteins). Considering

the robust complementation of the SDS / EDTA sensitivity phenotype, we expect the vancomycin resistance phenotype is more complex, and may be informed by RNAseq studies.

Interestingly, although vancomycin resistance of *mfa* deletion mutants is widely accepted in the field, we do not find evidence of published complementation for this phenotype – nor, therefore, of a lack of complementation either. We wanted to be fully transparent and share this phenotype, and its lack of complementation, in case this has implications for our global understanding of the Mla system in the future.

- SDS/EDTA sensitivity assay: The concentrations of SDS reported (~0.003-0.004%) are ~100-fold lower than what is typically used for *E. coli mfa* mutants. We would like to confirm that these values are correct and not a typo.

Answer:

Thank you for checking - yes, these are not typos. *Vp* is much more sensitive to SDS than *E. coli*, which may reflect a difference in composition of its outer membrane. We also expect this difference in OM permeability may reflect the difference in niches between these two organisms; whilst *Vp* typically inhabits the oral tract, *E. coli* must adapt to the gastrointestinal tract by resisting the action of host detergents, i.e. bile salts.

- Have the authors tried predicting a homo-hexamer of the beta-barrel domain? If they are analogous to MlaA, they would need to create a pathway for lipids to move from within the OM to the tunnel through the MCE protein. It would be interesting if AF suggests they form a stable assembly, vs each barrel doing its own thing in isolation.

Answer:

We indeed tried to model homo-hexamers of the C-terminal beta-barrel domain alone. While AlphaFold-multimer predicts a tighter ring of six beta-barrels, the prediction confidence scores are very low (pTM of 0.279 and ipTM of 0.144 for the highest-ranked model); and the PAE is close to the maximum error (30 Å) between beta-barrels, so we do not consider this arrangement as a reliable prediction. This information is now provided in Supplementary Fig 16 below.

- In 71: Perhaps it would be appropriate to ref the earlier work establishing the IM complex here: Kamischke

2019, Ekiert 2017, and Thong 2016; if there is space, one could also cite the more recent 6 papers for the high resolution structures that all came out around the same time (PMIDs: 33845086, 34188171, 33298869, 33199922, 32884137, 33236984). In the context of the MlaA-Omp complex, we could also imagine citing Abellon-Ruiz 2017 here.

Answer:

Thank you, we have added some of the corresponding references. As the introduction initially stated (now line 62) "In *E. coli*, the Mla system is composed of..." we had only included structures / papers within this organism.

However, considering how crucial these articles are (which we do mention later in our manuscript) we have adapted this sentence to:

"In model diderms such as *E. coli* and *A. baumannii*, the Mla system is composed of..." and added the suggested references.

- Fig. 1 legend: The authors refer to the MCE protein as a "substrate binding protein", akin to MBP in a classical ABC importer. However, the MCE protein is playing a completely different role (if anything, MlaC would be analogous to a SBP, but that is absent in the *V. parvula* system). We suggest removing this terminology.

Answer:

Thank you, we have removed this terminology.

- Line 118-121: The authors mention making single *mlaDEF* mutations and that all *V. parvula mla* mutants display hypersensitivity to SDS/EDTA and increased resistance to vancomycin. However, not all of this data is shown, so this claim is not well supported. We suggest showing all the data or altering the claim to match the data shown.

Answer:

Thank you for this comment. As shown in Supp Fig 2, phenotypes between single and triple *mla* mutants are not additive for growth kinetics, LPS profiles, autoaggregation and biofilm formation. We have now adapted this supplementary figure to also explicitly show the SDS / vancomycin sensitivity of $\Delta mlaF$ (the only missing mutant, as $\Delta mlaE$ is not viable), as $\Delta mlaD$ and $\Delta mlaEFD$ are already presented in Main Fig 2.

- In 131: "these phenotypes are indicative of a loss of lipid asymmetry": We might suggest softening the claim here. This may be correct, but OM asymmetry has not been directly assessed, and this system appears to be quite different from *mla*. It may be remodeling the OM in a different way, and we think the observed phenotypes don't necessarily have to arise from a loss of asymmetry.

Answer:

We have softened the claim by modifying the sentence to:

"Together, these phenotypes are indicative of a remodeling of the OM and possibly a loss of lipid asymmetry".

- In 151-153: Is there any precedent for vinyl ether bonds in bacterial lipids? Perhaps a reference to prior work here would be appropriate, and a statement of how common/uncommon these are?

Answer:

Yes, plasmalogens (a type of ether lipid, which are lipids containing vinyl ether bonds as opposed to ester bonds) are typically restricted to anaerobic bacterial species. 2 references have been added. We have also added clarity that the presence of vinyl ether bonds is indicative of plasmalogens / ether lipids.

Lines 146-149: "NMR analyses also revealed that ~15-35% of each lipid species present in the envelope of *V. parvula* contains a vinyl ether bond (indicative of plasmalogens^{27,28}) which could result from the activity of the identified plasmalogen biosynthesis homologue".

- In 176-177: Are these 2 independent insertions at the exact same location?

Answer:

Yes, the two clones are identical for the Tn insertion in *tamB*. This has been clarified in the text. However, as indicated in (now) line 176, a spontaneous suppressor of $\Delta mlaD$ was also identified with a SNP encoding a STOP codon closer to the C-terminus of *tamB* (see Fig 3b).

- In 179-180: How many AsmA-like proteins are encoded in the genome? This may affect the claim made in In 196-198, which could perhaps be softened a bit to avoid being overly speculative.

Answer:

As we have answered to the reviewer above:

Yes, we did identify one other gene encoding a potential AsmA-like protein in Vp. We constructed all combinations of mutants of *mldD*, *tamB* and "*asmA*", and *asmA* does not appear to have a significant impact on OM homeostasis in the combinations of deletion that were tested. Indeed, even the triple $\Delta mldD\Delta tamB\Delta asmA$ mutant is viable, suggesting there are as yet unidentified PL trafficking systems in *V. parvula*. We also see that deletion of *asmA* in the *mldD* background does not suppress the associated OM permeability phenotype, nor that of the hypervesiculation.

Considering this gene was also not identified in any suppressor or transposon mutagenesis screens for the *mldD* mutant, we decided not to introduce supplementary information on it to simplify our message. However, we will likely use this information for a future study as we continue to probe the systems which govern PL homeostasis in our new model organism.

We have added this information:

Lines 362-363: "Indeed, one other *asmA*-like protein exists in *V. parvula*, and further work is required to elucidate its functional role, if any, in the context of OM homeostasis."

- In 203-204: Does that mean that MlaF has a long C-terminal tail that may "handshake" with the neighbouring MlaF subunit, even in the absence of MlaB? Or is the *V. parvula* MlaF a little shorter? Also, in other MlaE structures there is an N-terminal amphipathic helix that runs parallel to the plane of the membrane, in the cytoplasmic leaflet. I can see it in Ext. Data Fig. 3D for *E. coli* but can't pick it out for the *V. parvula* prediction. Is it missing in *V. parvula* MlaE? I think it would be worth 1 sentence commenting on these differences if they are indeed different.

Answer:

MlaF in *V. parvula* indeed has a C-terminal extension that "handshakes" with the other MlaF subunit in the predicted model, though with a lower associated confidence value compared to the rest of the protein. The extension is actually shorter in Vp compared to *E. coli* (14 vs 22 a.a.), whereas in *E. coli*, the last C-ter residues (absent in Vp) also interact with MlaB (absent in Vp). Regarding MlaE, we thank the reviewer for noticing the absence of the elbow helices. We actually used a shortened sequence of Vp MlaE for the AlphaFold prediction, which did not include the full N-terminal region. We have now performed the Alpha-Multimer prediction of the MlaEFD₁₋₁₃₀ complex using the full-length MlaE sequence. The new model is virtually identical to the previous one, except that the elbow helix is present, with the same orientation as in the *E. coli* complex, and the position of the MlaD TMD is closer to the one found in the *E. coli* complex structure, still with lower pLDDT values.

We have updated the corresponding now Supplementary Fig. 8 (former Ext. Fig 3) panels and added the following sentence in the revised manuscript:

"This MlaDEF_{vp} model includes both the N-terminal elbow helix in MlaE and a shorter version of the C-terminal extension of MlaF interacting with the neighboring MlaF subunit, despite the absence of an MlaB counterpart."

- In 281-283: We think mycobacteria are regarded by many in the field as diderm (though I have had heated debates about this with colleagues), while we agree that many other actinobacteria are monoderms. Cryo ET of mycobacteria reveals an OM that appears very much like the OM of Gram-negatives (e.g., Hoffman, PNAS, 2008). So we suggest removing the specific mention of *M. tuberculosis* as a monoderm.

Answer:

Thank you for this comment - we have removed the mention of *M. tuberculosis*. For clarity, we refer to the ancestor of Actinobacteria that has lost the classical OM; a new OM, the mycomembrane, has emerged in a clade to which *M. tuberculosis* belongs. We were referring to the phylum as being monoderm as it had initially lost the OM.

We have clarified this in lines 286-290: "Interestingly, while we found MlaDE proteins in Corynebacteriales (consistent with their function as cholesterol transporters through the mycobacterial outer membrane as recently shown (Chen 2023)), we also identified multiple copies of MlaDE homologues in monoderm members of Actinobacteria - mainly the orders Solirubrobacterales, Thermophilales, Pseudonocardiales - and whose function is unknown."

- Main Text Figures: The middle 3 Figures need to be re-labeled to match the in-text references (the second Fig. 1 → Fig. 2, the current Fig. 2 → Fig. 3 and the current Fig. 3 → Fig. 4).

Answer:

Sorry for the mistake. This has been corrected.

- Fig 3A: Are the strips for each strain from the same plate, or is this a composite image? In order to compare the suppressor mutants to the controls, they need to be on the same plate, but I'm not sure that this is the case?

Answer:

Yes, the suppressor mutants were plated on the same plate than the WT and *m1aD* mutant. Please see the information provided in the raw data for Figures file (below).

Fig 3: TamB is a suppressor of the $\Delta mlaD$ phenotype

Fig 3A: Serial dilution plating of all 10 Tn-insertion suppressor mutants of $\Delta mlaD$. As shown in the paper, only 4 Tn insertions substantially rescued the detergent sensitivity phenotype of $\Delta mlaD$. The red boxes highlight the area presented in Main Fig 3B.

- Fig. 3C: Please indicate in the legend what the dashed line box is highlighting (i.e. the hydrophobic regions that are predicted to be membrane-embedded). Also, perhaps this is due to the angle of the model, are there only three C-terminal beta barrels (reflecting the possibility that several beta-barrels can fuse to form a trimeric conformation, as mentioned in the Discussion) or are another three just hidden from view?

Answer:

Thank you – the dashed boxes indeed highlight the expected positions of the membrane embedded regions according to their hydrophobicity. This was added to the legend.

There are 6 beta barrels in Fig 4c (left) but the three in the background are hidden by the three in front. In the right view (cut-away), the three barrels in the foreground are removed (same for the MlaD TM helices in the IM).

- Fig. 5: We think it would be very helpful to have three different categories of "MlaD" proteins, and color code them differently (currently, they are all one shade): 1) true MlaD proteins (with short C-terminal helical region); 2) Single MCE domain proteins with a long helical region, with beta-barrels; and 3) Single MCE domain proteins with a long helical region, WITHOUT beta-barrels.

Answer:

Thank you for your comment – we have added a schematic to more clearly show the three versions of MlaD. This information is further detailed in Fig. 5 and in Supp Fig 12.

- Extended Fig.3A: Label the figure with "i" and "ii" as written in the legend. Also, the legend mentions that the major lipid species in *V. parvula* are labeled 1-4 in the figure, however the #4 in Aii looks like it runs similarly to the cardiolipin standard. This is a little confusing as the text mentions that there is no cardiolipin detected by MALDI-QIF-TOF.

Answer:

We think the reviewer refers to supplementary Fig. S3 - these labels have been corrected.

Concerning band #4, the migration of this band in Supp Fig. 3ai clearly shows that it does not migrate like cardiolipin. In S3a_{ii}, the ratio of chloroform/methanol/water is changed as compared to S3a_i, and this is a coincidence that the migration looks like cardiolipin. We also see coincidental migrations of lipids of another

model we are studying, *D. radiodurans*, with PE, PC and CL even though this species has been described not to contain any of these lipids. While some important extra work is needed to identify band #2 and band #4, we know these bands are not cardiolipin and this is compatible with the fact that we did not identify any Vp genes with homology to genes encoding cardiolipin synthases in other organisms. The cardiolipin standard remains a valuable addition to the TLCs when different solvent mixtures are used and provoke different migration patterns.

- Extended Fig. 3Bi: The sentence “The relative decrease in these two species from the mla KO OMVs confirmed the relative enrichment of PE” is a little confusing. Is there a typo here or can the authors elaborate more on the connection between the reduction of unknown lipid and PE increase.

Answer:

Sorry for the confusion. We do not know whether there is a connection between these unknown lipids and PE. The assumption we are making is just regarding the relative quantity of these different lipids. We see that the quantity of other lipids does not seem to change (somehow behaving as internal controls of equivalent loading), and since these unknown lipids are reduced while PE is enhanced, this statement is just to indicate that the increased quantity in PE is not due to an overloading of the samples. We have removed this statement to avoid confusion, and just modified the previous sentence to: “PMA staining of lipid extracts from WT and Δmla OMVs shows a decrease in the relative quantities of two unknown lipid species, highlighted in white, as opposed to enhanced detection of PE”.

- Extended Fig. 5: In the legend, A, B, D, E refer to hexameric chains A to G. Should this not be A to F, as labeled in the figure? It would be more useful in understanding the figure if the legend detailed what each dashed line box was highlighting (around rank 4 of A, B, C and the AlphaFold models of C and F. Also, having the PAE colored error bar labeled with “high error” and “low error” as in Ext. Fig. 3C would be helpful (can only label one representative error bar so as to not clutter the figure).

Answer:

We corrected the labelling of chains in the legend of Extended Fig. 5 (now Supplementary Figure 10). Black dashed boxes indicate the position of an “open groove” when the first (A) and last (F) helices are not in contact. The blue dashed box indicates the AF-multimer model MlaD₃₆₋₂₆₃ that was used to build the MlaD_{Vp} full-length model. This information was added to the legend of Extended Fig. 5, and labels have been added for the PAE colour bar (now Supplementary Figure 10).

- Extended Fig. 6: “Presence OF absence of Mla components...” I believe should be “Presence OR absence”. We also cannot read the detailed species names, as well as for Ext. Fig. 7.

Answer:

Thank you, this typo has been corrected.

- In order to make the AlphaFold predictions readily accessible to the community, coordinates should either be deposited in ModelArchive or included as supplementary files as part of the publication.

Answer:

The following PDB files were added as supplementary material:

* Supplementary Dataset 2: Vp_MlaDEF_AF2.pdb : AlphaFold model of *V. parvula* MlaD₁₋₁₃₀EF with 6:2:2 stoichiometry

* Supplementary Dataset 3: Vp_MlaD_FL_hexamer_AF2.pdb : AlphaFold model of full-length *V. parvula* MlaD homo-hexamer

Reviewer #1 (Remarks to the Author):

In this work "Bridges instead of boats? The Mla system of diderm Firmicute *Veillonella parvula* reveals an ancestral transenvelope core of phospholipid trafficking", Grasekamp, et al. investigate inter membrane phospholipid trafficking in the diderm Firmicute, *Veillonella parvula* and identify a system with homology to part of the *E. coli* Mla system and determine that this system functions in retrograde phospholipid transport using a somewhat different mechanism than *E. coli*, mediated by a trans envelope architecture of MlaD. This revision thoroughly addresses previous concerns I had related to clarity. The manuscript is well written and should be of interest to a broad audience.

Reviewer #2 (Remarks to the Author):

The authors have improved their manuscript and addressed concerns appropriately. This study was very well done, the data are clear, and its conclusions are justified. This work significantly advances our understanding of the cell envelope in bacteria and clearly illustrates the need to investigate non-model bacteria. The manuscript should be of interest to a broad readership.

Reviewer #3 (Remarks to the Author):

The authors have adequately addressed essentially all of my comments. I am still not crazy about this system being called Mla, as in my opinion it is quite distinct, and that is part of what makes it so exciting! But I have said my piece and will leave it to the authors to decide what they would like to call their discovery. Really nice manuscript!